# PROXY-FDA: PROXY-BASED FEATURE DISTRIBUTION ALIGNMENT FOR FINE-TUNING VISION FOUNDATION MODELS WITHOUT FORGETTING

## ABSTRACT

Vision foundation models pre-trained on massive data encode rich representations of real-world concepts, which can be adapted to downstream tasks by fine-tuning. However, fine-tuning foundation models on one task often leads to the issue of *concept forgetting* on other tasks, and this issue is exacerbated by the typically limited data for fine-tuning. Recent methods of robust fine-tuning aim to mitigate forgetting of prior knowledge without affecting the fine-tuning performance. Knowledge is often preserved by matching the original and fine-tuned model weights or feature pairs. However, such point-wise matching can be too strong, without explicit awareness of the feature neighborhood structures that encode rich knowledge as well. We propose a novel regularization method **Proxy-FDA** that explicitly preserves the structural knowledge in feature space. Proxy-FDA performs Feature Distribution Alignment (using nearest neighbor graphs) between the pre-trained and fine-tuned feature spaces, and the alignment is further improved by informative proxies that are generated dynamically to increase data diversity. We show in end-to-end fine-tuning experiments that Proxy-FDA significantly reduces concept forgetting, and we find a strong correlation between forgetting and a distributional distance metric (in comparison to L2 distance). We further demonstrate Proxy-FDA's utility in both few-shot (based on prompt tuning) and continual fine-tuning settings, where we achieve consistent gains over the corresponding baselines.

## 1 INTRODUCTION

Vision foundation models like CLIP (Radford et al., 2021) and DINOv2 (Oquab et al., 2024) pre-trained on large amounts of data demonstrate remarkable performance across various tasks and data distributions. Such foundation models are known to have learned vast knowledge on real-world concepts that can serve as a useful prior for downstream task adaptation via fine-tuning. Existing fine-tuning methods include end-to-end finetuning, linear probing, prompt tuning (Zhou et al., 2022a;b), and adapter learning (Gao et al., 2021). While these methods prove effective, empirical evidence shows that they frequently suffer from an undesirable effect called *concept forgetting* (Mukhoti et al., 2024). Forgetting occurs when a fine-tuned model overfits on the downstream task, and unlike its pre-trained counterpart, significantly loses the ability to recognize concepts on other tasks. The issue is even more pronounced when limited data size and diversity are available for fine-tuning.

Concept forgetting has driven recent research on robust fine-tuning with the goal of preserving the pre-trained knowledge *and* performing well on the downstream task. One simple approach is to ensemble models before and after fine-tuning (Wortsman et al., 2022b). Alternative methods are based on regularization techniques to constrain the fine-tuned model to remain close to the original foundation model in either weight space (Li et al., 2018) or feature space (Mukhoti et al., 2024). Feature-space regularization by matching the pre-trained and fine-tuned features across samples shows a more promising effect in reducing forgetting, since it directly minimizes the change in input-output behaviour of the model. One key assumption behind such regularization is that the L2 feature-space distance is a good indicator of the similarity of encoded concepts in different models.

We argue that aligning individual feature points imposes too strong of a constraint. Without an explicit insight of feature neighborhoods, the concepts preserved *point-wise* are found to be limited, resulting

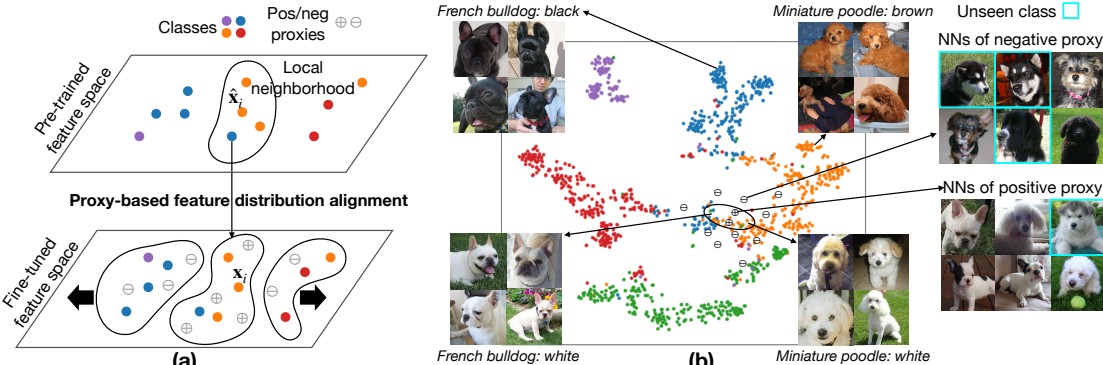

Figure 1: **(a)** Illustration of Proxy-based Feature Distribution Alignment (Proxy-FDA), where the pre-trained and fine-tuned feature distributions are aligned by their local neighborhood structures, which is further aided by proxies (*i.e.*, synthetic features). Proxy-FDA offers a structure-wise feature regularization to preserve the rich knowledge in neighborhood structures. We show Proxy-FDA significantly outperforms point-wise feature regularization in alleviating concept forgetting during fine-tuning. **(b)** t-SNE visualization of the local feature neighborhood (circled) on ImageNet for the pre-trained CLIP ViT-B/16 model. In this neighborhood, we observe the same white color from two dog breeds "French bulldog" and "Miniature poodle". Preserving CLIP's common-sense knowledge (in this case the color attribute shared across different classes) using FDA maintains the generalizability of foundation models. On the other hand, the generated proxies include diverse information from both seen and unseen (*e.g.*, "Malamute") classes that can regularize the neighborhood boundary and further improve FDA. The synthesized seen/unseen class data are illustrated by kNN retrieval from the base/new class splits of ImageNet when fine-tuning on the base only.

in sub-optimal performance of forgetting reduction. Here we suggest that it is desirable to explicitly inform the fine-tuning process of the local structure of feature neighborhoods. By preserving this neighborhood structure with a *structure-wise* regularization term, the rich knowledge encoded in the local structure of the original feature space will be transferred to the fine-tuned one. As a result, the fine-tuned model can forget significantly less while still maintaining its downstream performance.

The above idea motivates us to propose a new feature regularization term called **Feature Distribution Alignment** (**FDA**). Specifically, we first model the structural relations of pre-trained features using a nearest neighbor graph. Then we transfer the graph to the fine-tuned feature space, where feature neighbors are pulled together while non-neighbors are pushed away regardless of their labels. Such FDA process enables sharing knowledge beyond class concepts in local feature neighborhoods, such as visual attributes or co-occurring patterns. Fig. 1(b) provides an example of the white color attribute (of two dog breeds) mined from a local neighborhood on ImageNet. This example represents the common-sense prior knowledge embedded in a vision foundation model that is often richer than the class labels on downstream datasets. Preserving such knowledge (*e.g.*, about color) during fine-tuning is important to maintain the generalizability of the foundation model, which can facilitate recognizing unfamiliar classes from different tasks. What is harmful is to just specialize on the task at hand, since all information (*e.g.*, color sensitivity) but its class label will be discarded.

Another key contribution of this paper is an improvement to FDA, with the introduction of a new regularization called **Proxy-FDA**, which uses *proxies* as synthetic features. This full method is particularly useful on data-deficient fine-tuning tasks (such as few-shot ones), where the limited task data do not allow sufficient alignment of complex feature distributions. To further increase data diversity, Proxy-FDA learns to generate a set of instance-wise proxies both within and outside a target feature's local neighborhood. Fig. 1(b) exemplifies some proxies that synthesize informative unseen data or unseen class concepts. We empirically show that the generated proxies improves FDA with richer data/concepts, thereby further reducing concept forgetting.

We have conducted experiments of fine-tuning vision foundation models end-to-end on ten classification tasks. Results show that Proxy-FDA significantly outperforms other fine-tuning baselines in preventing concept forgetting, without hurting the downstream accuracy. We also find a strong correlation between concept forgetting and a distance metric OTDD – Optimal Transport Dataset Distance (Alvarez-Melis & Fusi, 2020) which is ideal to measure the alignment quality for feature distributions with local structures. Crucially, the correlation between concept forgetting and the

structure-aware OTDD metric indicates the need of structure-wise FDA in some form for better forgetting mitigation. Our structure-wise Proxy-FDA is shown to indeed forget much less than point-wise feature regularization (Mukhoti et al., 2024). We further show Proxy-FDA can be plugged into various prompt tuning methods to perform few-shot fine-tuning, where Proxy-FDA shows consistent gains and superior data efficiency for lowering forgetting. Lastly, Proxy-FDA proves effective on continual fine-tuning tasks, outperforming specialized continual learning baselines as well.

In summary, our **main contributions** include:

- A novel regularization method, Proxy-FDA, that aligns the local structures of feature distributions with learned proxies, aiming to preserve concepts when fine-tuning vision foundation models;
- Correlation analysis between concept forgetting and a structure-aware distributional distance metric, OTDD, which implicitly explains the success of our structure-wise FDA method;
- State-of-the-art performance on reducing concept forgetting in three settings: end-to-end, few-shot (based on prompt tuning) and continual fine-tuning.

## 2 RELATED WORK

**Robust fine-tuning.** End-to-end fine-tuning often suffers from concept forgetting and degraded out-of-distribution (OOD) performance. In the foundation model era, linear probing or that followed by end-to-end tuning (Kumar et al., 2022) are common remedies to maintain the OOD robustness of a pre-trained model. Alternative methods either ensemble the original and fine-tuned models (Wortsman et al., 2022b;a) or use the contrastive pre-training loss directly for fine-tuning (Goyal et al., 2023). More recently, Song et al. (2023) propose a method called FD-Align, which trains a spurious feature classifier and maintains its output consistency during fine-tuning. As a result, FD-Align significantly improves the OOD accuracy. To prevent forgetting, regularization methods are often used to minimize the model distance before and after fine-tuning in either weight space (Li et al., 2018) or feature space (Mukhoti et al., 2024). In few-shot settings, regularization is even more important. For example, the prompt learning method CLIPood (Shu et al., 2023) regularizes via temporal model ensembling, while PromptSRC (Khattak et al., 2023b) directly regularizes the output features and logits between pre-trained and prompt-tuned models. Nevertheless, all existing methods do not explicitly account for feature neighborhood structures, which we show is key for robust fine-tuning.

**Feature and data distribution alignment.** These techniques have been explored in different contexts. At the core of measuring **distributional distances**, Optimal Transport (OT) (Villani, 2008) provides a principled approach to compare data distributions in a geometrically meaningful way. Given the similar nature of our FDA method that aligns the "clustering" structures of distributions, we use an OT-based distance metric OTDD (Alvarez-Melis & Fusi, 2020) to measure FDA quality. Feature alignment is also key to **Domain Adaptation** (DA) (Wang & Deng, 2018). However, most DA methods learn a separate domain-invariant feature subspace to align domains implicitly, which differs from our explicit FDA during fine-tuning. More related to our method is the **Knowledge Distillation** (KD) field (Wang & Yoon, 2021), where traditional KD methods match features or probability distributions between teacher and student models. Relation-based KD methods are particularly similar to our high-level idea by distilling feature relations in form of kNNs (Zhu et al., 2022), feature similarities (Park et al., 2019; Passalis & Tefas, 2018; Tung & Mori, 2019; Peng et al., 2019) and relative ranks (Chen et al., 2018). Our Proxy-FDA can be seen as an alternative relational KD method that distills both kNNs and similarities, and further improves with proxy learning.

**Proxy learning.** This approach is widely adopted in deep metric learning (Movshovitz-Attias et al., 2017; Kim et al., 2020; Roth et al., 2022) to reduce the sampling complexity of pure sample-based methods. Proxies are learned as class prototypes to optimize sample-proxy distances in place of sample-sample distances, resulting in faster convergence. By contrast, our proxy learning is different in both implementation and motivation: we learn instance-wise proxies via adaptive pooling of true samples; we also do not use the proxies as sample stand-ins, but as rich augmentations for improving FDA. This makes our approach more related to those **feature augmentation** methods, such as by random linear interpolation (Verma et al., 2019) and outlier feature synthesis (Du et al., 2022; Tao et al., 2023). Empirically, our method is more effective than these feature augmentation methods by generating diverse augmented features from the entire feature neighborhood.

## 3 METHOD

We aim for forgetting-free fine-tuning of vision foundation models (*e.g.*, CLIP and DINOv2), using feature-space regularization based on *Feature Distribution Alignment* (FDA). Specifically, given a pre-trained model $f_{\hat{\theta}}$, we use the downstream dataset $\mathcal{D}_{\text{ft}}$ to fine-tune the model into $f_{\theta}$. Our goal is to specialize the fine-tuned model on $\mathcal{D}_{\text{ft}}$ with low task loss $\mathcal{L}_{\text{task}}$ (*e.g.*, cross-entropy loss for classification), whilst preventing concept forgetting on any target dataset $\mathcal{D} \neq \mathcal{D}_{\text{ft}}$. To prevent forgetting, we introduce an FDA-based regularization term to the downstream task loss, which gives:

$$\mathcal{L} = \frac{1}{B} \sum_{i=1}^{B} \left( \mathcal{L}_{\text{task}}^{i} + \lambda \mathcal{L}_{\text{FDA}}^{i} \right), \quad (1)$$

where $\mathcal{L}_{\text{FDA}}^{i}$ is the FDA loss for each sample $i$ in a mini-batch $\{i\}_{i=1}^{B}$ of size $B$, and $\lambda$ is a weighting parameter.

### 3.1 FEATURE DISTRIBUTION ALIGNMENT (FDA)

Having defined the learning problem and its general loss function, we now present our FDA method in detail. During fine-tuning on $\mathcal{D}_{\text{ft}}$, we first use the pre-trained model $f_{\hat{\theta}}$ and fine-tuned $f_{\theta}$ to extract batch features $\hat{X} \in \mathbb{R}^{d \times B}$ and $X \in \mathbb{R}^{d \times B}$, respectively. Note $\hat{X} = [\hat{x}_1, \ldots, \hat{x}_B]$ are the pre-trained batch features with $\hat{x}_i \in \mathbb{R}^d$, while $X = [x_1, \ldots, x_B]$ are the features currently being fine-tuned with $x_i \in \mathbb{R}^d$. To transfer the structural knowledge in $\hat{X}$ into $X$, we align the structural relations of $\hat{X}$ and $X$ based on their nearest neighbor graphs.

Concretely, for each pre-trained feature point $\hat{x}_i$, we maintain its $k$-nearest neighbor set $R_i = \{j | \hat{x}_j \in \text{kNN}(\hat{x}_i)\}$ within the batch. Note $|R_i| = K$, and we will detail later how to construct batches to facilitate the kNN search. This way, we obtain an instance-wise batch partition from the pre-trained model's perspective, leading to the positive set of neighbors $\hat{X}_i^{+} = \hat{X}(R_i) \in \mathbb{R}^{d \times K}$ and negative set of non-neighbors $\hat{X}_i^{-} \in \mathbb{R}^{d \times (B-K-1)}$. To form the complete nearest neighbor graph, we further compute the cosine similarities between pre-trained features $\hat{w}_{ij} = \cos(\hat{x}_i, \hat{x}_j)$ for $j \in \{1, \ldots, B\}$ and $j \neq i$. Accordingly, we organize them into similarity vectors for neighbors $\hat{w}_i^{+} \in \mathbb{R}^K$ and non-neighbors $\hat{w}_i^{-} \in \mathbb{R}^{B-K-1}$.

For efficient graph matching between $\hat{X}$ and $X$, we choose to simply transfer the neighbor indices $R_i$ and similarities $\{\hat{w}_i^{+}, \hat{w}_i^{-}\}$ from $\hat{X}$ to $X$. This means neighbors in the pre-trained feature space should remain neighbors in the fine-tuned feature space. Hence among $X$, we similarly have a positive set $X_i^{+} = X(R_i) \in \mathbb{R}^{d \times K}$ where the identified neighbors are pulled together in the fine-tuned feature space, and a negative set $X_i^{-} \in \mathbb{R}^{d \times (B-K-1)}$ where non-neighbors are pushed away. On the other hand, we associate the pre-trained feature similarities $\{\hat{w}_i^{+}, \hat{w}_i^{-}\}$ with $\{X_i^{+}, X_i^{-}\}$ to preserve fine-grained feature neighborhood structures. We will show that transferring both the neighbor indices and similarities works better than only transferring the former, *i.e.*, binary partitioning of neighbors vs. non-neighbors. Fig. 2(a) and (c) – the FDA parts – visualize the high-level idea.

To capture the desired structures, we use the Sigmoid loss, which is shown by (Zhai et al., 2023) to be a noise-resistant objective to handle a variable number of positives and negatives per batch:

$$\mathcal{L}_{\text{FDA}}^{i} \left( \{X_i^{+}, X_i^{-}\}, \{\hat{w}_i^{+}, \hat{w}_i^{-}\} \right) = \frac{1}{(|X|-1)} \sum_{x_j \in X, j \neq i} \log \left( 1 + e^{w_{ij} \left( -\frac{\cos(x_i, x_j)}{\tau} + b \right)} \right), \quad (2)$$

where $w_{ij}$ is a weighting parameter. $w_{ij}$ equals $\hat{w}_{ij}$ if $j \in R_i$ (*i.e.*, weighting by $\hat{w}_i^{+}$ for neighbors), and $-\hat{w}_{ij}$ if $j \notin R_i$ (*i.e.*, weighting by $-\hat{w}_i^{-}$ for non-neighbors). $\tau$ and $b$ are learnable parameters which are initialized in a similar way as in (Zhai et al., 2023). The above FDA loss helps to preserve local neighborhood structures in the fine-tuned feature space, without involving class labels.

**Batch construction and neighborhood size $K$.** To have a meaningful characterization *and* alignment of local neighborhood structures, we need to ensure that each mini-batch has diverse class distributions that may overlap locally in the feature space, and that a sufficient number of neighbors $|R_i| = K$ are identified for each sample in a batch.

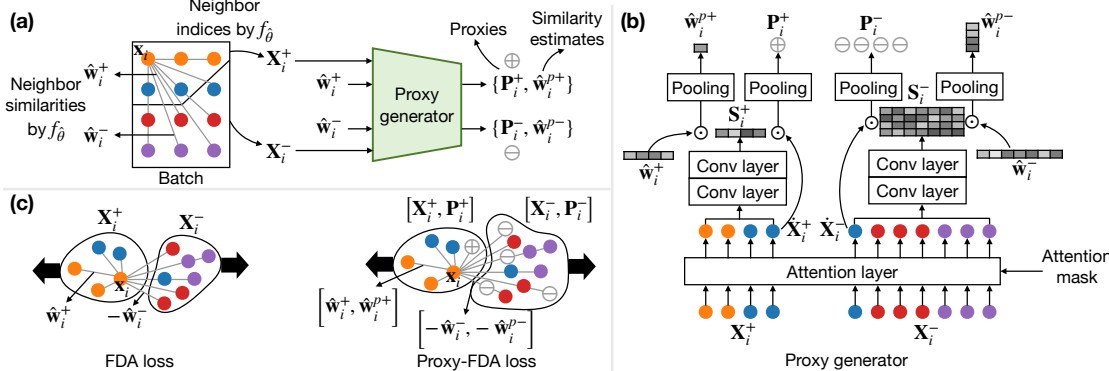

Figure 2: **(a)** Overview of our batch construction, transfer of the nearest neighbor graph of pre-trained model $f_{\hat{\theta}}$ (with neighbor indices and similarities), and proxy generator. **(b)** Efficient architecture of the proxy generator that generates dynamic proxies or synthetic features (details in Appendix B). **(c)** (Proxy-) FDA loss penalizes local distribution overlap between the similarity-weighted positive $\boldsymbol{X}_i^+$ and negative features $\boldsymbol{X}_i^-$, (with) and without using proxies $\{\boldsymbol{P}_i^+, \boldsymbol{P}_i^-\}$.

To meet the above-mentioned requirements, we *sample batch data in a class-balanced manner*, with $n$ samples for each of the $m$ classes. For a fixed batch size $B = m \cdot n$ that best fits in the available GPU memory, we choose a high value of $m$ to increase the diversity of class concepts in batch, but at the cost of reducing the number of examples per class $n$. By default, $m = 16$ and $n = 4$. More critically, we perform **hard class mining** to construct batches where samples from different classes are similar (details in Appendix A). This enables meaningful kNN search within a batch.

*For the neighborhood size $K$, we choose $K > n$ to guarantee that there is more than one class in any identified local feature neighborhood $R_i$. This way, each neighborhood includes an adaptive selection of "small clusters" from related classes. FDA between such neighborhoods will encourage transferring high-level knowledge beyond class concepts. Fig. 1(b) exemplifies a common color attribute mined locally for two similar dog classes. Preserving this knowledge that is embedded in foundation models is important to prevent forgetting during fine-tuning. Note it is possible that the inter-class similarity is not high enough in $R_i$ (thus relatively low $\hat{w}_{ij}$ for inter-class samples and no shared properties between classes), then FDA reverts back to aligning class semantics.*

## 3.2 PROXY-FDA

One challenge with FDA is that the downstream dataset $\mathcal{D}_{\text{ft}}$ can be limited in both data size and diversity. In this case $\mathcal{D}_{\text{ft}}$ does not allow adequate FDA, thereby preserving only limited concepts from those learned during pre-training. To address the data challenge, one could retrieve external data assuming the pre-training dataset is often inaccessible. However, using external data will inevitably suffer from higher compute/memory cost as well as various levels of distributional shift. Here we propose a compute- and data-efficient approach to improve downstream data diversity and eventually improve FDA quality. Our approach involves generating synthetic features or *proxies* on-the-fly from observed fine-tuning data. Such generated proxies have no distributional shift since they adapt to the considered feature distribution. We leave sampling suitable external data for FDA to future work.

Before diving into the details of Proxy-FDA, we highlight two points. First, for each feature point $\boldsymbol{x}_i$ in a batch, both its positive set $\boldsymbol{X}_i^+$ and negative set $\boldsymbol{X}_i^-$ could lack diversity. Second, the semi-hard nature of $\boldsymbol{X}_i^-$ (due to hard class mining) alleviates the "limited diversity" issue. This is because semi-hard negatives can provide the most informative signal for shaping the decision boundary, without sampling the vast space of negative features.

To further increase data diversity, we learn to generate two-sets of proxies $\boldsymbol{P}_i^+ = [\boldsymbol{p}_1^+, \ldots, \boldsymbol{p}_{n^{p+}}^+] \in \mathbb{R}^{d \times n^{p+}}$ and $\boldsymbol{P}_i^- = [\boldsymbol{p}_1^-, \ldots, \boldsymbol{p}_{n^{p-}}^-] \in \mathbb{R}^{d \times n^{p-}}$ out of $\boldsymbol{X}_i^+ \in \mathbb{R}^{d \times K}$ and $\boldsymbol{X}_i^- \in \mathbb{R}^{d \times (B-K-1)}$ respectively. $n^{p+}$ and $n^{p-}$ are made proportional to the size of $\boldsymbol{X}_i^+$ and $\boldsymbol{X}_i^-$ using a scalar $s$, see details in Appendix D. The proxies are learned to be as diverse as possible but still lie in the corresponding true feature manifold. This way $\boldsymbol{P}_i^-$ is still semi-hard, just as $\boldsymbol{X}_i^-$. Fig. 1(b) shows that both $\boldsymbol{P}_i^+$ and $\boldsymbol{P}_i^-$ can synthesize unseen data/concepts. Such unseen information will provide fine-grained regularization of the neighborhood boundary, and will improve FDA with richer concepts.

Following the above intuitions, we define our proxy learning loss $\mathcal{L}^i_{\text{proxy}} = \mathcal{L}_{\boldsymbol{P}^+_i} + \mathcal{L}_{\boldsymbol{P}^-_i}$, where:

$$\mathcal{L}_{\boldsymbol{P}^+_i} = \frac{1}{n^{p+}} \sum_{j=1}^{n^{p+}} \frac{1}{|\boldsymbol{X}|} \sum_{\boldsymbol{x}_l \in \boldsymbol{X}} \log\left(1 + e^{w_l\left(-\frac{\cos(\boldsymbol{p}^+_j, \boldsymbol{x}_l)}{\tau} + b\right)}\right) + \alpha \cdot \mathcal{L}_{\text{var}}(\boldsymbol{P}^+_i), \tag{3}$$

$$\mathcal{L}_{\boldsymbol{P}^-_i} = \frac{1}{n^{p-}} \sum_{j=1}^{n^{p-}} \frac{1}{|\boldsymbol{X}|} \sum_{\boldsymbol{x}_l \in \boldsymbol{X}} \log\left(1 + e^{w_l\left(-\frac{\cos(\boldsymbol{p}^-_j, \boldsymbol{x}_l)}{\tau} + b\right)}\right) + \alpha \cdot \mathcal{L}_{\text{var}}(\boldsymbol{P}^-_i). \tag{4}$$

The first loss term constrains proxies $\boldsymbol{P}^+_i$ and $\boldsymbol{P}^-_i$ towards the feature manifolds $\boldsymbol{X}^+_i$ and $\boldsymbol{X}^-_i$. This is achieved using the binary label $w_l$ which, in case of $\mathcal{L}_{\boldsymbol{P}^+_i}$, equals 1 if $\boldsymbol{x}_l \in \boldsymbol{X}^+_i$ and -1 if $\boldsymbol{x}_l \in \boldsymbol{X}^-_i$; while in case of $\mathcal{L}_{\boldsymbol{P}^-_i}$, is the opposite. The variance loss $\mathcal{L}_{\text{var}}(\boldsymbol{P})$ maximizes proxy diversity in form of $1/d \sum_{j=1}^d \max(0, 1 - \sqrt{\text{Var}(\boldsymbol{P}_{j,:}) + \epsilon})$ with $\epsilon$ being a small scalar. $\alpha$ is a weighting parameter.

In practice, we use Eq. (3-4) to train our proxy generator online during the model fine-tuning process. This ensures the generated proxies always adapt to the current feature distribution. Appendix B details the **network architecture of the proxy generator**, which only incurs a small compute cost as shown in Appendix D. At high level, conditioned on $\boldsymbol{X}^+_i$ and $\boldsymbol{X}^-_i$, our proxy generator is trained to predict the instance-wise proxies $\{\boldsymbol{P}^+_i, \boldsymbol{P}^-_i\}$ and their similarity estimates $\{\hat{\boldsymbol{w}}^{p+}_i, \hat{\boldsymbol{w}}^{p-}_i\}$ all at once. Finally, we use all the predictions to augment the true features $\{\boldsymbol{X}^+_i, \boldsymbol{X}^-_i\}$ and similarities $\{\hat{\boldsymbol{w}}^+_i, \hat{\boldsymbol{w}}^-_i\}$, arriving at our Proxy-FDA loss for feature-space regularization:

$$\mathcal{L}^i_{\text{Proxy-FDA}} = \mathcal{L}^i_{\text{FDA}}\left(\left\{[\boldsymbol{X}^+_i, \boldsymbol{P}^+_i], [\boldsymbol{X}^-_i, \boldsymbol{P}^-_i]\right\}, \left\{[\hat{\boldsymbol{w}}^+_i, \hat{\boldsymbol{w}}^{p+}_i], [\hat{\boldsymbol{w}}^-_i, \hat{\boldsymbol{w}}^{p-}_i]\right\}\right),$$

$$\mathcal{L} = \frac{1}{B} \sum_{i=1}^B \left(\mathcal{L}^i_{\text{task}} + \lambda \mathcal{L}^i_{\text{Proxy-FDA}}\right). \tag{5}$$

## 4 EXPERIMENTS

In this section, we benchmark concept forgetting and methods addressing it. Appendix D includes analysis of the hyper-parameters of our Proxy-FDA method, and Appendix E ablates the key components of Proxy-FDA. Here we perform three groups of experiments: 1) End-to-end fine-tuning of different vision foundation models on 10 image classification datasets, 2) Parameter-efficient fine-tuning via prompt tuning on 11 classification datasets in few-shot settings, often with severe forgetting issues, 3) Continual fine-tuning on a sequence of classification tasks as another stress test.

### 4.1 END-TO-END FINE-TUNING

**Datasets.** We follow (Mukhoti et al., 2024) to use 10 image classification datasets: Stanford Cars (Krause et al., 2013), CIFAR-10/100 (Krizhevsky, 2009), DTD (Cimpoi et al., 2014), EuroSAT (Helber et al., 2019), GTSRB (Stallkamp et al., 2012), MNIST (LeCun et al., 2010), RESISC45 (Cheng et al., 2017), SVHN (Netzer et al., 2011) and ImageNet (Deng et al., 2009). These datasets include various semantic concepts, thus are perfect to benchmark forgetting of the rich concepts that may have already been learned during pre-training.

**Setting and baselines.** The image encoder of CLIP model (ViT-B/32) is fine-tuned end-to-end on the 10 datasets. We compare with popular end-to-end fine-tuning methods all using the cross-entropy loss as $\mathcal{L}_{\text{task}}$. The baselines include naive fine-tuning and LP-FT methods (Kumar et al., 2022). They differ in the linear head initialization, with zero-shot weights (text encodings of class name) and Linear Probe (LP) weights respectively. While L2SP (Li et al., 2018) and LDIFS (Mukhoti et al., 2024) add a point-wise regularization between the original and fine-tuned models in weight- and feature-space respectively. By contrast, our (Proxy-)FDA imposes a structure-wise regularization in feature space. Note except for the naive fine-tuning baseline, LP initialization is used for all methods including ours for a fair comparison of different regularization techniques.

**Metrics.** When fine-tuning on dataset $\mathcal{D}_{\text{ft}}$, we report two evaluation metrics: LP accuracy $\mathcal{A}_{\text{LP}}$ on the test set of $\mathcal{D}_{\text{ft}}$ (*i.e.*, the fine-tuning performance itself), and the change $\Delta_{\text{LP}}$ in $\mathcal{A}_{\text{LP}}$ between

Table 1: **Test accuracy $\mathcal{A}_{\text{LP}}$ of end-to-end fine-tuned model on each dataset and its average $\Delta_{\text{LP}}$ computed over other datasets**. The image encoder of CLIP ViT-B/32 is used here. $\Delta_{\text{LP}}$ denotes the change in $\mathcal{A}_{\text{LP}}$ between pre-trained and fine-tuned models on target dataset $\mathcal{D}$, quantifying the level of concept forgetting. Higher $\Delta_{\text{LP}}$ shows lower forgetting or positive forward transfer ($\Delta_{\text{LP}} > 0$).

| Dataset | Naive End-to-End | | LP-FT | | L2SP | | LDIFS | | FDA (ours) | | Proxy-FDA (ours) | |
|---|---|---|---|---|---|---|---|---|---|---|---|---|
| | $\mathcal{A}_{\text{LP}}$ | $\Delta_{\text{LP}} \uparrow$ | $\mathcal{A}_{\text{LP}}$ | $\Delta_{\text{LP}} \uparrow$ | $\mathcal{A}_{\text{LP}}$ | $\Delta_{\text{LP}} \uparrow$ | $\mathcal{A}_{\text{LP}}$ | $\Delta_{\text{LP}} \uparrow$ | $\mathcal{A}_{\text{LP}}$ | $\Delta_{\text{LP}} \uparrow$ | $\mathcal{A}_{\text{LP}}$ | $\Delta_{\text{LP}} \uparrow$ |
| Cars | 83.48 | -1.56 | 84.95 | -0.63 | 83.87 | 0.47 | 85.26 | -0.18 | **85.36** | 1.02 | 84.69 | **1.26** |
| CIFAR10 | **97.73** | -1.60 | 97.71 | -0.81 | 97.66 | 1.16 | 97.24 | 1.18 | 97.53 | 1.55 | 97.61 | **1.63** |
| CIFAR100 | 88.60 | -0.96 | 88.41 | -0.11 | 86.94 | 1.03 | **88.99** | 0.86 | 88.21 | 1.44 | 88.33 | **1.51** |
| DTD | 77.18 | -3.01 | 72.18 | -1.76 | 74.63 | 0.01 | 75.27 | 0.53 | 77.22 | 1.04 | **77.28** | **1.19** |
| EuroSAT | 98.76 | -5.72 | **98.87** | -3.75 | 98.20 | -0.85 | 98.22 | 1.32 | 98.53 | 1.61 | 98.63 | **1.74** |
| GTSRB | 98.52 | -5.90 | **98.53** | -0.94 | 95.00 | 1.18 | 97.81 | 1.27 | 98.16 | 1.58 | 97.79 | **1.69** |
| MNIST | 99.67 | -8.76 | **99.68** | -6.02 | 99.18 | 1.49 | 99.52 | 2.64 | 99.43 | 2.76 | 99.49 | **2.81** |
| RESISC45 | **95.76** | -3.79 | 95.56 | -2.27 | 94.13 | 0.66 | 95.13 | 0.90 | 95.31 | 1.18 | 95.63 | **1.43** |
| SVHN | 97.30 | -11.12 | **97.50** | -8.73 | 96.54 | -2.11 | 96.95 | -0.29 | 96.96 | 0.67 | 96.65 | **0.92** |
| ImageNet | 82.02 | -1.26 | 82.12 | -0.87 | 80.78 | -0.10 | **82.21** | 0.35 | 81.93 | 1.05 | 82.16 | **1.22** |
| Mean across 10 datasets | **91.90** | -4.37 | 91.55 | -2.59 | 90.69 | 0.29 | 91.66 | 0.86 | 91.86 | 1.39 | 91.82 | **1.54** |

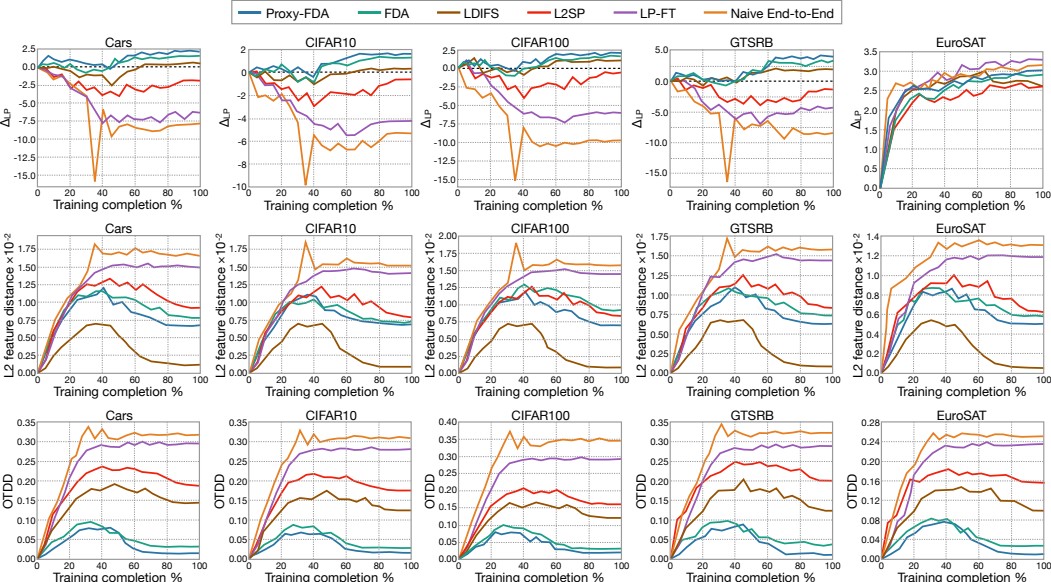

Figure 3: Three metrics computed over the course of model fine-tuning (CLIP ViT-B/32) on **EuroSAT**: $\Delta_{\text{LP}}$ (**Top row**), L2 feature-space distance (**Middle row**) and distributional distance metric OTDD (**Bottom row**), all between pre-trained and fine-tuned models. Our (Proxy-)FDA achieves the best results in preventing concept forgetting on other datasets (highest positive $\Delta_{\text{LP}}$) without hurting the downstream performance on EuroSAT. We also observe that concept forgetting measured by $\Delta_{\text{LP}}$ is more correlated to OTDD than L2 feature distance (see text for details).

pre-trained and fine-tuned models on a different dataset $\mathcal{D} \neq \mathcal{D}_{\text{ft}}$. Negative $\Delta_{\text{LP}}$ indicates **concept forgetting** on $\mathcal{D}$, while a positive value indicates **positive forward transfer**. Clearly, the higher $\Delta_{\text{LP}}$ the better. When $\mathcal{D} = \mathcal{D}_{\text{ft}}$, $\Delta_{\text{LP}}$ on $\mathcal{D}$ simply denotes the change of downstream performance, and we expect $\Delta_{\text{LP}}$ to increase over the course of fine-tuning.

To gain insights on what impacts the concept forgetting performance, we further monitor two distance metrics for distribution alignment during fine-tuning: point-wise L2 distance between pre-trained and fine-tuned feature pairs without using any distributional information, and Optimal Transport Dataset Distance (OTDD) (Alvarez-Melis & Fusi, 2020) that takes feature distribution structures into consideration (details in Appendix C). Between the two distance metrics, OTDD is generally more suited to measure the alignment quality for feature distributions with local structures as in our case.

**Results.** Table 1 compares $\mathcal{A}_{\text{LP}}$ on each fine-tuning dataset and the $\Delta_{\text{LP}}$ averaged over other datasets. We observe that FDA obtains a positive average $\Delta_{\text{LP}}$ for all fine-tuning tasks, thereby achieving a positive forward transfer. Proxy-FDA further improves the average $\Delta_{\text{LP}}$ consistently.

This is not the case for naive fine-tuning and LP-FT where the average $\Delta_{\text{LP}}$ is all negative indicating concept forgetting. Point-wise regularization methods L2SP and LDIFS obtain mostly positive $\Delta_{\text{LP}}$ but significantly lower than our results, highlighting the benefits of our structure-wise feature regularization and proxy feature generation.

We also observe that our good performance on forgetting prevention does not compromise (much) the downstream fine-tuning accuracy $\mathcal{A}_{\text{LP}}$. The mean $\mathcal{A}_{\text{LP}}$ (across 10 datasets) of (Proxy-)FDA is (91.82) 91.86, which is only slightly lower than that of naive fine-tuning 91.90 but outperforms all other results. Overall, our structure-wise regularization method achieves the best trade-off between concept forgetting and downstream performance. Fig. 3 (top row) exemplifies the fine-tuning task on EuroSAT, where (Proxy-)FDA consistently outperforms other baselines in forgetting prevention during fine-tuning (higher $\Delta_{\text{LP}}$), but has similar performance on EuroSAT in the meantime.

Fig. 3 (middle and bottom rows) shows how L2 feature distance and OTDD change as we fine-tune EuroSAT using different methods. Overall, both the distance metrics are correlated to concept forgetting — fine-tuning methods with smaller L2 distance/OTDD forget less with higher $\Delta_{\text{LP}}$, while methods with a larger distance suffer more from forgetting with lower $\Delta_{\text{LP}}$. The only exception to the overall trend is when we use L2 feature distance to compare (Proxy-)FDA with LDIFS. We see that (Proxy-)FDA, while having larger L2 distance than LDIFS, still forgets less. On the contrary, (Proxy-)FDA consistently gets lower OTDD. This suggests that **the structure-aware OTDD is a better indicator of concept forgetting compared to the point-wise L2 distance**. More crucially, the fact that OTDD is more correlated to forgetting than L2 distance reaffirms that having some form of structure-wise FDA can mitigate forgetting better. Finally, we note our (Proxy-)FDA is only applied on EuroSAT samples, but the mitigation of forgetting extends to all other datasets. This indicates the generalizing effect of our feature regularization method, which can preserve pre-trained knowledge without requiring third party datasets during fine-tuning.

Table 4 in Appendix shows that our benefits still hold when end-to-end fine-tuning happens with different architectures of CLIP (Radford et al., 2021), FLAVA (Singh et al., 2022), DINOv2 (Oquab et al., 2024) and MAE (He et al., 2022). Proxy-FDA consistently provides the highest $\Delta_{\text{LP}}$ values across foundation models and architectures, achieving positive forward transfer in all cases. Proxy-FDA also achieves the best $\mathcal{A}_{\text{LP}}$ in many cases, which is encouraging.

### 4.2 FEW-SHOT PROMPT TUNING

**Datasets.** We follow (Zhou et al., 2022b) to use 11 datasets, again with a wide range of visual concepts and domains: ImageNet (Deng et al., 2009), Caltech101 (Fei-Fei et al., 2004), Oxford-Pets (Parkhi et al., 2012), StanfordCars (Krause et al., 2013), Flowers102 (Nilsback & Zisserman, 2008), Food101 (Bossard et al., 2014), FGVC-Aircraft (Maji et al., 2013), SUN397 (Xiao et al., 2010), DTD (Cimpoi et al., 2014), EuroSAT (Helber et al., 2019) and UCF101 (Soomro et al., 2012).

**Settings and metrics.** Prompt tuning is adopted for parameter-efficient fine-tuning in the few-shot scenario. We consider the two settings introduced in (Zhou et al., 2022b): 1) **Base-to-new class generalization** within each dataset, *i.e.*, prompt tuning on the base class split as $\mathcal{D}_{\text{ft}}$, and evaluating on the disjoint base and new class splits to obtain $\mathcal{A}_{\text{Base}}$ and $\mathcal{A}_{\text{New}}$. To quantify concept forgetting on the unseen new class split, we further report $\Delta_{\text{New}}$ as the change in $\mathcal{A}_{\text{New}}$ between pre-trained and prompt-tuned models – the higher $\Delta_{\text{New}}$ the lower forgetting. 2) **Cross-dataset generalization** with ImageNet for prompt tuning and other 10 datasets for evaluation. Similarly, we report both the test accuracy $\mathcal{A}$ and accuracy change $\Delta_{\mathcal{A}}$ on each dataset to quantify forgetting. The cross-dataset setting is more challenging due to the presence of both domain- and class-incremental distribution shifts, *e.g.*, from fine-grained flowers classification to satellite imagery recognition on EuroSAT. For all experiments, we report results as an average over three random seeds.

**Implementation.** We apply our Proxy-FDA regularization to different prompt tuning baselines. For fair comparisons, we use the same implementation details of each baseline, including the prompt length, learning rate schedule and tuning epochs for each dataset. By default, all methods use 16 shots per class to prompt tune the CLIP model (Radford et al., 2021) with ViT-B/16.

**Results.** In Table 2, we report results in the base-to-new setting. Proxy-FDA is applied to two categories of methods: 1) regularization-free prompt tuning baselines, which learn text prompts

Table 2: **Few-shot prompt tuning in the base-to-new class generalization setting** (16 shots per class). $\mathcal{A}_H$ denotes the Harmonic mean of $\mathcal{A}_{Base}$ and $\mathcal{A}_{New}$. $\Delta_{New}$ denotes the change in $\mathcal{A}_{New}$ between pre-trained and prompt-tuned CLIP models. Higher $\Delta_{New}$ shows lower level of concept forgetting on the new class split of the considered dataset. On average, our Proxy-FDA consistently improves $\Delta_{New}$ for all prompt tuning methods, with competitive $\mathcal{A}_{Base}$ at the same time. Full results in Table 5.

| | | Prompt tuning without regularization | | | | | | | | Regularization-based | | | |
| | | CoOp | | CoCoOp | | VPT | | MaPLe | | CLIPood | | PromptSRC | |
| +Proxy-FDA | | ✗ | ✓ | ✗ | ✓ | ✗ | ✓ | ✗ | ✓ | ✗ | ✓ | ✗ | ✓ |
|---|---|---|---|---|---|---|---|---|---|---|---|---|---|
| | $\mathcal{A}_{Base}$ | 82.69 | **83.16** | **80.47** | 80.36 | **81.61** | 81.55 | 82.28 | **82.74** | 83.91 | **84.33** | 84.26 | **84.47** |
| Avg across | $\mathcal{A}_{New}$ | 63.22 | **73.67** | 71.69 | **76.44** | 69.61 | **73.89** | 75.14 | **77.13** | 74.50 | **76.54** | 76.10 | **77.45** |
| 11 datasets | $\Delta_{New}$ ↑ | -10.99 | **-0.55** | -2.53 | **2.22** | -4.61 | **-0.33** | 0.92 | **2.91** | 0.28 | **2.33** | 1.88 | **3.23** |
| | $\mathcal{A}_H$ | 71.66 | **78.13** | 75.83 | **78.35** | 75.14 | **77.53** | 78.55 | **79.84** | 78.93 | **80.25** | 79.97 | **80.81** |

Figure 4: **(a-b)** The average $\Delta_{New}$ with varying number of shots per class for prompt tuning in the base-to-new setting. FDA achieves higher gains over the baselines in low-data regime, and our proxy learning further improves data efficiency. **(c)** PromptSRC+Proxy-FDA scales better with data than end-to-end fine-tuning and its improved variants (FD-Align and WiSE-FT) in the few-shot setting.

(CoOp (Zhou et al., 2022a), CoCoOp (Zhou et al., 2022b)), image prompts (VPT (Jia et al., 2022)) or both (MaPLe (Khattak et al., 2023a)). 2) regularization-based prompt learners. CLIPood (Shu et al., 2023) maintains a weighted ensemble of the pre-trained and fine-tuned models. State-of-the-art PromptSRC (Khattak et al., 2023b) combines the ensembling strategy with both feature- and logit-level regularization between the original and fine-tuned models (but in a point-wise manner).

We can see from Table 2 that, averaged across 11 datasets, Proxy-FDA consistently improves the $\mathcal{A}_{New}$ of all regularization-free baselines, sometimes by a large margin (10.45 for CoOp), with competitive $\mathcal{A}_{Base}$ at the same time. The gains in $\mathcal{A}_{New}$ translate to gains in $\Delta_{New}$, indicating the utility of Proxy-FDA in lowering forgetting for few-shot settings. The per-dataset results in Table 5 (in Appendix) show that $\Delta_{New}$ sees particularly large gains on 3 semantically distant datasets (DTD, EuroSAT and UCF101), thanks to our strong capability of preserving pre-trained knowledge. Overall, Proxy-FDA boosts the $\mathcal{A}_H$ of MaPLe to 79.84, being already better than or on par with that of the regularization methods CLIPood (78.93) and PromptSRC (79.97). Encouragingly, Proxy-FDA is complementary to the two regularization methods and can further improve them in all metrics.

Apart from the above observations, recall that one of our motivations for Proxy-FDA is to help data-scarce downstream tasks, where even parameter-efficient prompt tuning can suffer from significant overfitting and forgetting. In Fig. 4(a-b), we vary the amount of data for prompt tuning and find that the $\Delta_{New}$ gain of FDA increases with less data. Meanwhile, our proxy learning component further improves data efficiency, often matching the FDA performance on half the data. Fig. 4(c) also shows the benefits of (Proxy-)FDA over end-to-end fine-tuning and its improved variants — FD-Align (Song et al., 2023) and WiSE-FT (Wortsman et al., 2022b) — in data-limited regimes.

Table 6 in Appendix compares results under the cross-dataset generalization setting. We plug Proxy-FDA into 3 representative baselines with and without regularization. Proxy-FDA is shown to prevent concept forgetting for all the 3 baselines, with uniformly increased $\Delta_{\mathcal{A}}$ on target datasets and a good trade-off with $\mathcal{A}$ on the source dataset.

## 4.3 CONTINUAL FINE-TUNING

Finally, we apply our approach to continual fine-tuning and see whether we can learn a sequence of downstream tasks without forgetting concepts. We follow the setup in (Mukhoti et al., 2024) to train on three task sequences: SVHN→CIFAR10→RESISC45, SVHN→CIFAR100→RESISC45 and

Table 3: **Continual fine-tuning: test accuracy $\mathcal{A}_{\text{LP}}$ and $\Delta_{\text{LP}}$ for models fine-tuned on three task sequences**. The first 3 rows show performance on fine-tuned tasks and the 4th row shows performance averaged on 6 other datasets.

| Fine-tune dataset | Evaluation dataset | Naive End-to-End | | LP-FT | | L2SP | | LDIFS | | FDA (ours) | | Proxy-FDA (ours) | |
|---|---|---|---|---|---|---|---|---|---|---|---|---|---|
| | | $\mathcal{A}_{\text{LP}}$ | $\Delta_{\text{LP}}\uparrow$ | $\mathcal{A}_{\text{LP}}$ | $\Delta_{\text{LP}}\uparrow$ | $\mathcal{A}_{\text{LP}}$ | $\Delta_{\text{LP}}\uparrow$ | $\mathcal{A}_{\text{LP}}$ | $\Delta_{\text{LP}}\uparrow$ | $\mathcal{A}_{\text{LP}}$ | $\Delta_{\text{LP}}\uparrow$ | $\mathcal{A}_{\text{LP}}$ | $\Delta_{\text{LP}}\uparrow$ |
| SVHN→ CIFAR10→ RESISC45 | SVHN | 90.29 | -7.13 | 90.97 | -6.46 | 91.93 | -4.53 | 96.68 | -0.41 | **96.77** | 0.61 | 96.72 | **0.93** |
| | CIFAR10 | 95.25 | -2.31 | 96.31 | -1.57 | 97.26 | -0.25 | **97.41** | -0.21 | 97.13 | 0.57 | 97.29 | **1.02** |
| | RESISC45 | 95.30 | 4.00 | 94.29 | 2.98 | 93.44 | 2.16 | 95.00 | 3.70 | 95.22 | 4.14 | **95.38** | **4.22** |
| | Others | 80.91 | -5.08 | 82.13 | -4.24 | 86.89 | -0.01 | 87.08 | 0.10 | **87.21** | 0.76 | 86.95 | **1.08** |
| SVHN→ CIFAR100→ RESISC45 | SVHN | 90.05 | -7.28 | 94.42 | -2.73 | 90.42 | -6.12 | 96.32 | -0.65 | 96.18 | 0.63 | **96.43** | **0.71** |
| | CIFAR100 | 81.08 | -7.18 | 82.63 | -3.04 | 85.72 | -0.88 | **86.54** | -0.30 | 86.33 | 0.72 | 86.14 | **0.85** |
| | RESISC45 | 95.40 | **4.13** | 93.81 | 2.51 | 93.21 | 1.90 | 95.11 | 3.83 | 95.32 | 3.95 | **95.46** | 4.01 |
| | Others | 83.76 | -4.65 | 85.14 | -4.02 | 89.04 | -0.37 | **89.12** | -0.23 | 89.02 | 0.68 | 89.09 | **0.96** |
| SVHN→ Cars→ RESISC45 | SVHN | 95.93 | -1.45 | 96.58 | -0.76 | 95.98 | -0.44 | 96.90 | -0.17 | 96.74 | 0.79 | **96.91** | **0.94** |
| | Cars | 76.96 | -4.18 | 71.60 | -8.36 | 81.82 | -0.40 | 84.23 | 0.47 | **84.38** | 1.14 | 84.32 | **1.36** |
| | RESISC45 | 95.17 | 3.89 | 94.35 | 3.00 | 93.43 | 2.13 | **95.27** | 3.73 | 95.12 | 3.92 | 95.23 | **4.07** |
| | Others | 83.38 | -4.93 | 84.39 | -4.51 | 87.15 | -1.67 | 89.39 | 0.23 | 89.54 | 0.96 | **89.67** | **1.17** |

SVHN→Cars→RESISC45. Table 3 shows our FDA and Proxy-FDA methods progressively improve the $\Delta_{\text{LP}}$ for each task sequence, both achieving positive forward transfer with all positive $\Delta_{\text{LP}}$ values. Proxy-FDA always attains the highest $\Delta_{\text{LP}}$ values (except on RESISC45 in the second sequence), while still remaining competitive in $\mathcal{A}_{\text{LP}}$. Table 8 and 9 in Appendix also show our benefits over popular continual learning baselines for both the 3-task setup and the classic class-incremental setting on Split ImageNet-R (Wang et al., 2022a).

## 5 CONCLUSION

This paper introduces Proxy-FDA, a novel feature-space regularization method that preserves concepts during fine-tuning. The core idea is to align the local structures of pre-trained and fine-tuned feature distributions with learned proxies. A structure-aware distributional distance metric is used to assess the feature alignment quality, demonstrating a strong correlation with concept forgetting. Our approach achieves state-of-the-art results in mitigating concept forgetting across end-to-end, few-shot, and continual fine-tuning settings.

**Limitations and future work.** We mainly study the forgetting of "concepts" at the granularity of categorical class labels. The class concepts are used for both method development (*e.g.*, class-balanced batch construction, and FDA across classes) and performance evaluation (on downstream classification datasets). To explore concepts beyond class labels, we could use natural language texts that have rich concepts at varying granularity. This requires different design choices for FDA-style methods to tackle the fine-grained concepts. As another line of future works, we plan to investigate whether Proxy-FDA can reduce forgetting when fine-tuning for other foundation model families like Large Language Models, or across different types of tasks like image segmentation and detection.

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

## A    HARD CLASS MINING

As mentioned in the main text (Section 3.1), we perform hard class mining in the mini-batch to facilitate the modeling and alignment of local neighborhood structures. The high-level idea of hard class mining is to greedily select class distributions that are close to one another. More specifically, we construct our mini-batch in the following way:

1. Randomly choose a large number of classes $C \gg m$; for each class, randomly sample $n$ examples to extract their feature embeddings using both $f_{\hat{\theta}}$ and $f_{\theta}$.
2. Sample a seed class randomly from the $C$ classes. Then greedily add a new class that has the largest class-wise loss $\sum_{i=1}^{n} \mathcal{L}_{\text{FDA}}^{i}$ (Eq. (2)) *w.r.t.* the selected classes till we reach $m$ classes. Note in this greedy process, we set the neighborhood size $K = n$ when computing $\mathcal{L}_{\text{FDA}}^{i}$.
3. Construct batch with the selected $m$ classes, each with $n$ examples.

## B    EFFICIENT ARCHITECTURE OF INSTANCE-WISE PROXY GENERATOR

Fig. 2(b) shows the network architecture of our proxy generator that is trained online using Eq. (3-4). The input $\boldsymbol{X}_i^+$ and $\boldsymbol{X}_i^-$ first go through an attention layer to model the global context within each set and fuse features thoroughly. Attention mask is used to ensure the independence between the two sets. Next, we dynamically pool the intermediate features $\dot{\boldsymbol{X}}_i^+ \in \mathbb{R}^{d \times K}$ and $\dot{\boldsymbol{X}}_i^- \in \mathbb{R}^{d \times (B-K-1)}$ via learned pooling functions, as summarized below. Through such pooling, we can predict proxies $\{\boldsymbol{P}_i^+, \boldsymbol{P}_i^-\}$ and their similarity estimates $\{\hat{\boldsymbol{w}}_i^{p+}, \hat{\boldsymbol{w}}_i^{p-}\}$ all at once.

$$\textbf{Predict pooling weights:} \quad \boldsymbol{S}_i^+ = h^+(\dot{\boldsymbol{X}}_i^+) \in \mathbb{R}^{K \times n^{p+}}, \quad \boldsymbol{S}_i^- = h^-(\dot{\boldsymbol{X}}_i^-) \in \mathbb{R}^{(B-K-1) \times n^{p-}}, \tag{6}$$

$$\textbf{Pooling in matrix form:} \quad \boldsymbol{P}_i^+ = \dot{\boldsymbol{X}}_i^+ \cdot \boldsymbol{S}_i^+ \in \mathbb{R}^{d \times n^{p+}}, \quad \boldsymbol{P}_i^- = \dot{\boldsymbol{X}}_i^- \cdot \boldsymbol{S}_i^- \in \mathbb{R}^{d \times n^{p-}}, \tag{7}$$

$$\hat{\boldsymbol{w}}_i^{p+} = \boldsymbol{S}_i^{+T} \cdot \hat{\boldsymbol{w}}_i^+ \in \mathbb{R}^{n^{p+}}, \quad \hat{\boldsymbol{w}}_i^{p-} = \boldsymbol{S}_i^{-T} \cdot \hat{\boldsymbol{w}}_i^- \in \mathbb{R}^{n^{p-}}, \tag{8}$$

$$\text{where} \qquad\qquad \hat{\boldsymbol{w}}_i^+ \in \mathbb{R}^K, \qquad\qquad \hat{\boldsymbol{w}}_i^- \in \mathbb{R}^{B-K-1}.$$

Note both $h^+(\cdot)$ and $h^-(\cdot)$ are implemented by two convolutional layers, but with different output channel sizes ($n^{p+}$ and $n^{p-}$ respectively). The output pooling weights $\boldsymbol{S}_i^+$ and $\boldsymbol{S}_i^-$ are softmax-normalized, leading to convex combinations of features and feature similarities during the pooling stage. This eases training of pooling functions and makes sure the pooled results are valid (especially the pooled similarity estimates).

## C    DISTRIBUTIONAL DISTANCE METRIC: OTDD

To measure FDA quality, there are many distance metrics for distribution alignment. Here we choose the distributional distance metric based on Optimal Transport Dataset Distance (OTDD) (Alvarez-Melis & Fusi, 2020). OTDD is especially suited to measure the alignment quality of feature distributions with local structures, because this distance metric takes both the label distribution and clustering structure of the feature distributions into consideration.

Specifically, OTDD uses the feature and label distributions $(\boldsymbol{x}, y)|_{\boldsymbol{x} \in \mathcal{X}, y \in \mathcal{Y}}$ to compute the distance between two datasets. Given that the source and target datasets may have different label sets, the high-level idea of OTDD is to represent each class label as a distribution over the in-class features. This transforms the source and target label sets into the shared space of feature distributions over $\mathcal{X}$. In our context of model fine-tuning, we have pre-trained features $\hat{\boldsymbol{x}}$ and fine-tuned features $\boldsymbol{x}$ that are likely shifted from $\hat{\boldsymbol{x}}$. They form the source and target feature distributions respectively, and have different labels $\hat{y}$ and $y$ (details later). Then we can define the label distance $D_{\mathcal{Y}}(\hat{y}, y)$ using the $p$-Wasserstein distance associated with the L2 distance $\|\hat{\boldsymbol{x}} - \boldsymbol{x}\|_2^2$ in $\mathcal{X}$. This enables one to measure the distributional difference in $\mathcal{X} \times \mathcal{Y}$:

$$D_{\mathcal{X} \times \mathcal{Y}}((\hat{\boldsymbol{x}}, \hat{y}), (\boldsymbol{x}, y)) = (D_{\mathcal{X}}(\hat{\boldsymbol{x}} - \boldsymbol{x})^p + D_{\mathcal{Y}}(\hat{y}, y)^p)^{1/p}. \tag{9}$$

Please refer to (Alvarez-Melis & Fusi, 2020) for the exact formulation. To capture the clustering structure of both the pre-trained and fine-tuned feature distributions, we perform K-Means clustering per class on each feature distribution. This results in pseudolabels $\hat{y}$ and $y$ that are more fine-grained than class labels for OTDD computation.

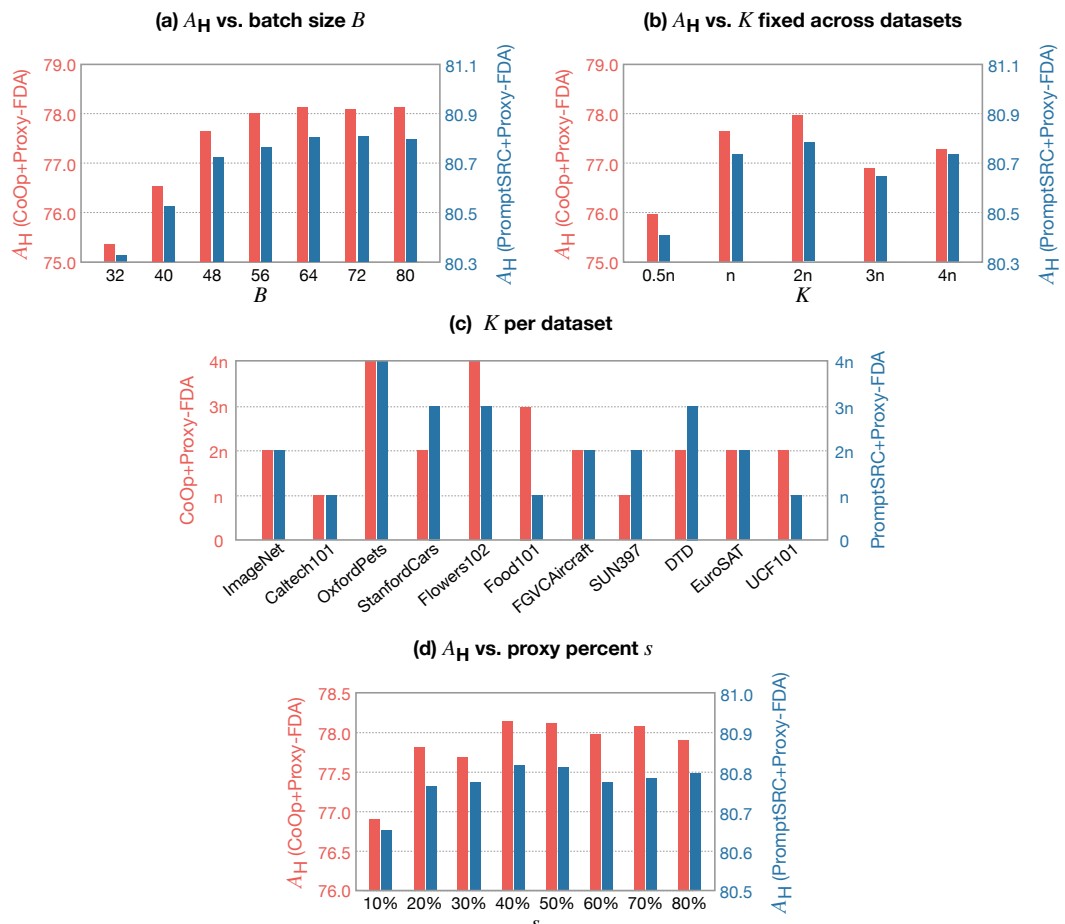

Figure 5: **Sensitivity analysis for hyper-parameters: (a)** batch size $B$, **(b)** neighborhood size $K$ that is fixed across datasets, **(c)** optimal $K$ per dataset, and **(d)** scalar $s$ that decides the percent number of generated proxies compared to that of real samples. Analysis is performed for few-shot prompt tuning in the base-to-new setting (16 shots per class). We report the $\mathcal{A}_{\mathrm{H}}$ averaged across 11 datasets, when applying Proxy-FDA to two representative baselines CoOp and PromptSRC. Note $\mathcal{A}_{\mathrm{H}}$ is the Harmonic mean of $\mathcal{A}_{\mathrm{Base}}$ (representing prompt-tuning accuracy itself) and $\mathcal{A}_{\mathrm{New}}$ (representing generalization and can derive $\Delta_{\mathrm{New}}$). Hence $\mathcal{A}_{\mathrm{H}}$ is ideal for hyper-parameter sweeping since $\mathcal{A}_{\mathrm{H}}$ denotes a trade-off between downstream accuracy and concept forgetting ($\Delta_{\mathrm{New}}$).

## D  ANALYSIS OF HYPER-PARAMETERS AND COMPUTE COST

**Hyper-parameters.**    Fig. 5(a) shows our Proxy-FDA approach benefits from a relatively large batch size $B$ to preserve meaningful structures of feature neighborhoods. Performance decreases when $B < 64$; when $B$ grows larger than 64, performance seems quite robust to varying batch size. By default, we set $B = 64$ that best fits in our GPU memory.

Based on the hard class mining strategy (Section A), we construct a mini-batch with $m = 16$ hard-mined classes, each with $n = 4$ class samples. Note in few-shot settings, each class may not have enough data ($< 4$) for sampling, *e.g.*, only 1 or 2 shots are available per class. In this case, we perform random data augmentation to guarantee $n = 4$ samples per class. On the other hand, a relatively large $m$ ensures diverse class distributions in a batch, which allows better characterization of local feature neighborhoods. Diverse classes also allow pooling rich proxies from them, resulting in unseen data variations or new class concepts to further improve FDA.

Our Proxy-FDA method has two key hyper-parameters: the neighborhood size $K > n$ and a scalar $s$. The latter makes the number of positive proxies $n^{p+} = s \cdot K$ and negative proxies $n^{p-} = s \cdot (B - K - 1)$ proportional to the set size of the true positives $\boldsymbol{X}_i^+ \in \mathbb{R}^{d \times K}$ and true negatives $\boldsymbol{X}_i^- \in \mathbb{R}^{d \times (B-K-1)}$.

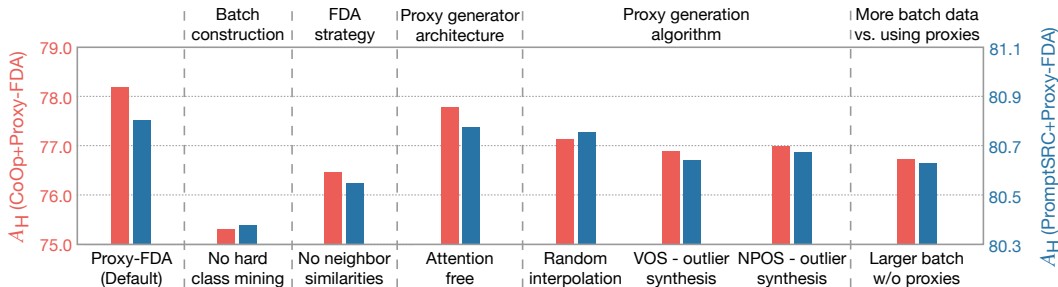

Figure 6: **Ablating the Proxy-FDA components** based on few-shot prompt tuning in the base-to-new setting (16 shots per class). We report the $\mathcal{A}_H$ averaged across 11 datasets, when applying Proxy-FDA to two representative baselines CoOp and PromptSRC. Note $\mathcal{A}_H$ is the Harmonic mean of $\mathcal{A}_{Base}$ (representing prompt-tuning accuracy itself) and $\mathcal{A}_{New}$ (representing generalization and can derive $\Delta_{New}$). Hence $\mathcal{A}_H$ is ideal for ablation studies since $\mathcal{A}_H$ denotes a trade-off between downstream accuracy and concept forgetting ($\Delta_{New}$). For the proxy generation strategy, we compare with random linear interpolation (Verma et al., 2019) and outlier feature synthesis methods VOS (Du et al., 2022) and NPOS (Tao et al., 2023).

The intuition of setting $K > n$ is to identify sufficient neighbors from more than one class, for meaningful FDA between similar clusters of related classes. Nevertheless, the exact value of $K$ is varied as a function of dataset distribution, as each dataset has different levels of intra- and inter-class variation. In practice, we pick the best $K$ per dataset from $\{n, 2n, 3n, 4n\}$. Fig. 5(b) shows how performance generally varies with $K$ when $K$ is fixed across 11 datasets. We see that $K = 2n$ works best, while it noticeably hurts performance when $K < n$, confirming our intuition above. Hence we stick to the constraint of $K > n$ for per-dataset $K$ selection (Fig. 5(c)).

On the other hand, the scalar $s$ is set to 0.4 by default. This leads to a virtual batch size of around 90 (increased from 64). The virtual batch now consists of true and synthetic features for FDA. Fig. 5(d) shows the sensitivity analysis for $s$.

Lastly, the weighting parameter for $\mathcal{L}_{var}$ (Eq. (4)) is fixed at $\alpha = 5$ for all experiments. We observe no meaningful improvements via more careful tuning of $\alpha$. The weighting parameter $\lambda$ is used to balance the task loss against our regularization loss (Eq. (1) and (5)). We tune $\lambda$ on a held-out validation set of each dataset.

**Compute cost.** Our Proxy-FDA mainly involves FDA and proxy generation. The proxy generator is lightweight with only one attention and two convolutional layers (totalling 23.6k parameters), which is negligible in comparison to the foundation model size. Here we show our feature regularization process only incurs a decent compute cost (on Nvidia A100 GPU). For end-to-end fine-tuning and few-shot prompt tuning tasks, averaged across the corresponding datasets, Proxy-FDA increases the fine-tuning time by 17% and 21% respectively, while FDA increases by 7% and 9%. Note Proxy-FDA does not impact the inference stage, hence we maintain the same FPS at the test time.

# E    ABLATING PROXY-FDA COMPONENTS

Fig. 6 includes ablation studies on the key components of Proxy-FDA, in the few-shot prompt tuning setting. We start with the batch construction strategy, comparing the default hard class mining method with random class sampling for a batch. Their considerable performance difference shows that hard class mining is crucial. Indeed, one can better model the nearest neighbor graphs from close class samples, which facilitates the following stage of graph alignment. Then what if we only align the neighbor indices between graphs, without considering the neighbor similarities (*i.e.*, keeping $\hat{w}_{ij} = 1$)? We see that this baseline leads to large performance drop as well, demonstrating that both neighbor indices and similarities are indispensable for FDA purpose.

Next, we isolate the impact of our proxy generator, in terms of both its architecture and training algorithm. For the architecture, we first note that our proxy generator is learned to produce unseen data out of diverse feature combinations within the positive set $\boldsymbol{X}_i^+$ or negative set $\boldsymbol{X}_i^-$. The attention

Table 4: **Test accuracy $\mathcal{A}_{\text{LP}}$ of end-to-end fine-tuned model on ImageNet and its average $\Delta_{\text{LP}}$ computed over 5 datasets (DTD, EuroSAT, GTSRB, RESISC45 and SVHN)**. We study different architectures of CLIP (Radford et al., 2021), FLAVA (Singh et al., 2022), DINOv2 (Oquab et al., 2024) and MAE (He et al., 2022). $\Delta_{\text{LP}}$ denotes the change in $\mathcal{A}_{\text{LP}}$ between pre-trained and fine-tuned models on target dataset, quantifying the level of concept forgetting. Higher $\Delta_{\text{LP}}$ shows lower forgetting or even positive forward transfer ($\Delta_{\text{LP}} > 0$). Note we initialize the model's linear head with zero-shot weights for naive fine-tuning, and with Linear Probe (LP) weights for all other methods including ours. The initialized zero-shot weights are the text encodings of class name for CLIP and FLAVA, and random weights for DINOv2 and MAE.

| Model | Architecture | Naive End-to-End | | LP-FT | | L2SP | | LDIFS | | FDA (ours) | | Proxy-FDA (ours) | |
|---|---|---|---|---|---|---|---|---|---|---|---|---|---|
| | | $\mathcal{A}_{\text{LP}}$ | $\Delta_{\text{LP}}\uparrow$ | $\mathcal{A}_{\text{LP}}$ | $\Delta_{\text{LP}}\uparrow$ | $\mathcal{A}_{\text{LP}}$ | $\Delta_{\text{LP}}\uparrow$ | $\mathcal{A}_{\text{LP}}$ | $\Delta_{\text{LP}}\uparrow$ | $\mathcal{A}_{\text{LP}}$ | $\Delta_{\text{LP}}\uparrow$ | $\mathcal{A}_{\text{LP}}$ | $\Delta_{\text{LP}}\uparrow$ |
| CLIP | ResNet-50 | 78.39 | -4.01 | 78.45 | -3.40 | 76.13 | -1.54 | 78.16 | -0.11 | 78.43 | 0.62 | **78.58** | **0.89** |
| | ViT-B/32 | 82.02 | -3.02 | 82.12 | -2.17 | 80.78 | -0.88 | **82.21** | 0.10 | 81.93 | 0.81 | 82.16 | **1.15** |
| | ViT-B/16 | 85.21 | -2.92 | 85.36 | -1.73 | 82.19 | -0.74 | 85.31 | 0.16 | **85.41** | 0.92 | 85.40 | **1.03** |
| | ViT-L/14 | 87.88 | -2.33 | 87.91 | -1.52 | 86.87 | -0.43 | 87.85 | 0.22 | **87.99** | 1.02 | 87.96 | **1.28** |
| FLAVA | ViT-B/16 | 81.18 | -3.94 | 81.36 | -3.04 | 80.11 | -1.10 | **81.61** | 0.04 | 81.47 | 0.61 | 81.59 | **0.96** |
| DINOv2 | ViT-B/14 | 85.32 | -2.71 | 85.48 | -1.86 | 84.50 | -0.66 | 86.02 | 0.06 | 86.23 | 0.68 | **86.34** | **0.85** |
| | ViT-L/14 | 87.60 | -1.92 | 87.90 | -1.40 | 87.02 | -0.19 | **87.91** | 0.13 | 87.87 | 0.77 | 87.71 | **0.94** |
| MAE | ViT-B/16 | 83.57 | -5.10 | 83.81 | -4.36 | 82.84 | -3.03 | 83.76 | -0.94 | 83.73 | -0.08 | **83.94** | **0.39** |
| | ViT-L/16 | 85.86 | -4.26 | **86.04** | -3.59 | 85.10 | -1.82 | 85.90 | -0.12 | 85.86 | 0.79 | 85.67 | **0.94** |

layer helps to achieve this goal by modeling the global context among all input features with pairwise attention. Convolutional layers, however, only have local receptive fields and have to rely on pooling operations to capture long-range dependencies. Here we compare with an attention-free architecture that has the attention layer replaced with convolutional plus pooling layers – the resulting proxy generator maintains a similar parameter count. The attention-free architecture is observed to achieve consistently lower performance, likely due to the lower quality of generated proxies.

Regarding the proxy generation algorithm, we compare with three baselines. One simple method is based on linear interpolation between random feature pairs from both $X_i^+$ and $X_i^-$. Feature similarity estimates are interpolated in the same way. Despite the simplicity, random interpolation obtains inferior performance than our learning-based approach — our approach can learn to synthesize more informative features to better help FDA. On the other hand, the parametric VOS and non-parametric NPOS methods learn to synthesize outlier features in low-likelihood regions (often around decision boundaries between classes). The two methods are observed to achieve even worse results than random interpolation. We conjecture that this is because outliers in low-likelihood regions are not able to encode diverse unseen data/concepts that are crucial for improving FDA.

To further quantify the effect of proxy learning that virtually increases the batch size $B$ from 64 to around 90, we compare with FDA simply on a larger batch with a similar number of true feature points. Specifically, we construct the batch with $m = 22$ hard-mined classes, each with $n = 4$ examples. Hence the batch size is comparable to that of Proxy-FDA, but without proxies. We observe from Fig. 6 that simply using a larger batch size does not perform as well. Instead, it is worth using our proxy generator to increase data diversity with only a small overhead.

## F  MORE RESULTS

**End-to-end fine-tuning.**  Table 4 shows the results of ImageNet fine-tuning with different foundation models and architectures. We see that both FDA and Proxy-FDA consistently improve the $\Delta_{\text{LP}}$ over other baselines, with Proxy-FDA offering the highest $\Delta_{\text{LP}}$ values. This comes with competitive downstream accuracy $\mathcal{A}_{\text{LP}}$ on ImageNet. Notably, our obtained $\Delta_{\text{LP}}$ values are mostly positive, with a sole exception of MAE model (ViT-B/16 architecture) when fine-tuned using FDA. This indicates that we can achieve positive forward transfer in most cases and otherwise minimized concept forgetting.

**Few-shot prompt-tuning.**  Table 5 lists the full results of prompt tuning on each of the 11 datasets under the base-to-new class generalization setting. Table 6 shows results under the cross-dataset generalization setting, *i.e.*, quantifying generalization from ImageNet to 10 target datasets. In both

settings, Proxy-FDA is plugged into different prompt tuning baselines. Proxy-FDA is observed to reduce concept forgetting consistently on unseen data with comparable performance on seen data.

We further compare with more recent prompt tuning methods in Table 7. Comparisons are conducted under the base-to-new class generalization setting, and an additional domain generalization setting. In the latter setting, we prompt tune on ImageNet (16 shots per class) and evaluate OOD generalization on ImageNetV2 (Recht et al., 2019), ImageNet-Sketch (Wang et al., 2019), ImageNet-A (Hendrycks et al., 2021b) and ImageNet-R (Hendrycks et al., 2021a) with different types of domain shift. The compared methods include ProText (khattak et al., 2024) and ArGue-N (Tian et al., 2024) that use LLMs to distill language priors into the learned prompts, as well as more related regularization methods OGEN (Zang et al., 2024) and CLAP (Lavoie et al., 2024). OGEN regularizes the prediction probabilities with an improved Mean Teacher, while CLAP regularizes the class prototypes (*i.e.*, class-wise feature means) for linear probing.

Table 7 shows that our structure-wise feature regularization method Proxy-FDA outperforms OGEN and CLAP in all metrics under the considered settings. Proxy-FDA achieves particularly large gains in generalization performance on the new classes or new domains, maximizing the positive forward transfer with higher $\Delta_{\text{New}}$. When compared to ProText and ArGue-N using external LLMs, our approach is LLM-free but achieves on-par or even better performance for both prompt-tuning and OOD generalization.

**Continual fine-tuning.** Table 8 compares our method with 5 classic continual learning methods in the 3-task setting: LwF (Li & Hoiem, 2017), LFL (Jung et al., 2016), iCaRL (Rebuffi et al., 2017), Distillation + Retrospection (D+R) (Hou et al., 2018) and ZSCL (Zheng et al., 2023).

Table 9 compares our method with recent continual learning methods on the class-incremental learning benchmark Split ImageNet-R. This benchmark divides the 200 classes from ImageNet-R into 10 tasks with 20 classes per task. The compared methods include LDIFS as well as L2P (Wang et al., 2022b), DualPrompt (Wang et al., 2022a), CODA-Prompt (Smith et al., 2023), Continual-CLIP (Thengane et al., 2022) and SLCA (Zhang et al., 2023). All methods use the same training (24,000) and testing (6,000) images. To further ensure fair comparisons, we follow the widely-adopted implementation: fine-tuning for 50 epochs using the Adam optimizer with $\beta_1 = 0.9$ and $\beta_2 = 0.999$. The initial learning rate is $1e^{-4}$, and we use a cosine learning rate scheduler as in (Mukhoti et al., 2024).

In both Table 8 and 9, our (Proxy-)FDA method outperforms all other methods in preventing forgetting. At the same time, (Proxy-)FDA is able to achieve the best fine-tuning performance.

**Knowledge distillation.** As metioned in the Related Work section, the high-level idea of our method resembles Knowledge Distillation (KD), epseically those relational KD methods that distill feature relations between models.

Table 10 shows our method is directly applicable to KD and quite performant. We follow the standard KD settings in (Zheng & Yang, 2024), and test teacher-student pairs using the same or different architectures of ResNet (He et al., 2016) and MobileNet (Howard et al., 2017) on ImageNet. We compare with state-of-the-art *logits matching* methods KD++ (Wang et al., 2023), DIST (Huang et al., 2022) and WTTM (Zheng & Yang, 2024). Note DIST can be viewed as a relational KD method at the logit level. We further compare with KD methods that *match feature relations* in form of kNNs (CNA (Zhu et al., 2022)) and feature similarities (ITRD (Miles et al., 2022)). CNA and ITRD are more related to our FDA method, but FDA differs in that both neighbor indices and similarities are distilled in the feature space. We see from the table that FDA consistently outperforms CNA and ITRD, and is competitive or better than logits-based DIST. Our proxy learning further improves performance, and Proxy-FDA is on par with the best prior work WTTM.

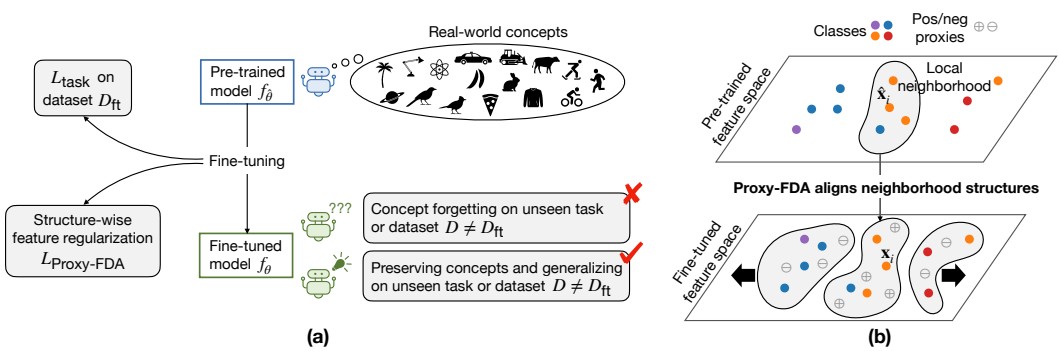

Figure 7: To replace Fig. 1(a) for a better motivation of our Proxy-FDA approach.

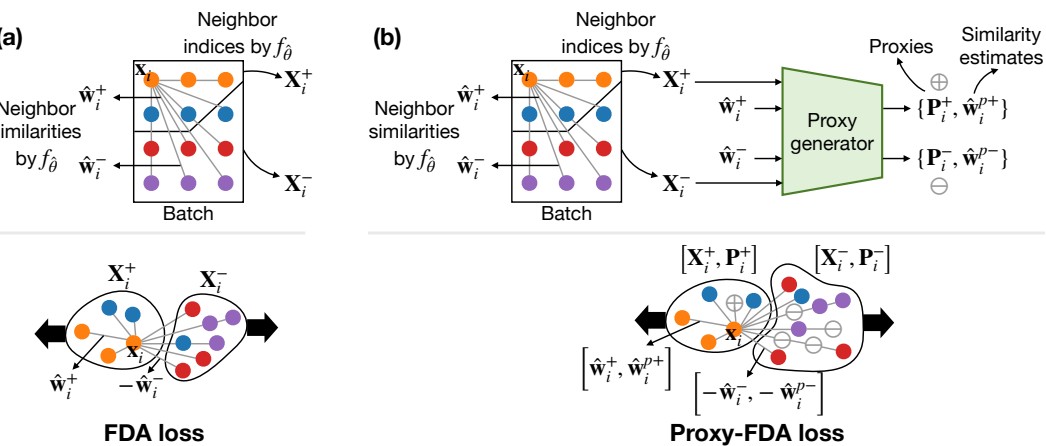

Figure 8: To replace Fig. 2(a)&(c) to only focus on the (Proxy-)FDA loss and improve interpretability.

Table 5: **Few-shot prompt tuning in the base-to-new class generalization setting** (16 shots per class). $\mathcal{A}_\text{H}$ denotes the Harmonic mean of $\mathcal{A}_\text{Base}$ and $\mathcal{A}_\text{New}$. $\Delta_\text{New}$ denotes the change in $\mathcal{A}_\text{New}$ between pre-trained and prompt-tuned CLIP models. Higher $\Delta_\text{New}$ shows lower level of concept forgetting on the new class split of the considered dataset.

| | | Prompt tuning without regularization | | | | | | | | Regularization-based | | | |
| | | CoOp | | CoCoOp | | VPT | | MaPLe | | CLIPood | | PromptSRC | |
| **+Proxy-FDA** | | ✗ | ✓ | ✗ | ✓ | ✗ | ✓ | ✗ | ✓ | ✗ | ✓ | ✗ | ✓ |
|---|---|---|---|---|---|---|---|---|---|---|---|---|---|
| **Avg across 11 datasets** | $\mathcal{A}_\text{Base}$ | 82.69 | **83.16** | **80.47** | 80.36 | **81.61** | 81.55 | 82.28 | **82.74** | 83.91 | **84.33** | 84.26 | **84.47** |
| | $\mathcal{A}_\text{New}$ | 63.22 | **73.67** | 71.69 | **76.44** | 69.61 | **73.89** | 75.14 | **77.13** | 74.50 | **76.54** | 76.10 | **77.45** |
| | $\Delta_\text{New}$ ↑ | -10.99 | **-0.55** | -2.53 | **2.22** | -4.61 | **-0.33** | 0.92 | **2.91** | 0.28 | **2.33** | 1.88 | **3.23** |
| | $\mathcal{A}_\text{H}$ | 71.66 | **78.13** | 75.83 | **78.35** | 75.14 | **77.53** | 78.55 | **79.84** | 78.93 | **80.25** | 79.97 | **80.81** |
| **ImageNet** | $\mathcal{A}_\text{Base}$ | 76.47 | 76.22 | 75.98 | 76.95 | 75.96 | 75.26 | 76.66 | 77.35 | 77.50 | 78.47 | 77.60 | 77.81 |
| | $\mathcal{A}_\text{New}$ | 67.88 | 72.97 | 70.43 | 73.48 | 67.32 | 71.25 | 70.54 | 71.51 | 70.30 | 72.07 | 70.73 | 71.55 |
| | $\Delta_\text{New}$ ↑ | -0.26 | 4.83 | 2.29 | 5.34 | -0.82 | 3.11 | 2.40 | 3.37 | 2.16 | 3.93 | 2.59 | 3.41 |
| | $\mathcal{A}_\text{H}$ | 71.92 | 74.56 | 73.10 | 75.17 | 71.38 | 73.20 | 73.47 | 74.32 | 73.72 | 75.13 | 74.01 | 74.55 |
| **Caltech101** | $\mathcal{A}_\text{Base}$ | 98.00 | 96.84 | 97.96 | 97.21 | 97.50 | 96.14 | 97.74 | 98.71 | 98.70 | 99.08 | 98.10 | 98.49 |
| | $\mathcal{A}_\text{New}$ | 89.81 | 97.45 | 93.81 | 97.15 | 94.10 | 95.93 | 94.36 | 95.42 | 94.60 | 95.01 | 94.03 | 95.34 |
| | $\Delta_\text{New}$ ↑ | -4.19 | 3.45 | -0.19 | 3.15 | 0.10 | 1.93 | 0.36 | 1.42 | 0.60 | 1.01 | 0.03 | 1.34 |
| | $\mathcal{A}_\text{H}$ | 93.73 | 97.14 | 95.84 | 97.18 | 95.77 | 96.03 | 96.02 | 97.04 | 96.61 | 97.00 | 96.02 | 96.89 |
| **OxfordPets** | $\mathcal{A}_\text{Base}$ | 93.67 | 95.01 | 95.20 | 96.96 | 96.05 | 95.32 | 95.43 | 95.42 | 95.70 | 97.63 | 95.33 | 96.31 |
| | $\mathcal{A}_\text{New}$ | 95.29 | 98.97 | 97.69 | 98.64 | 95.84 | 98.42 | 97.76 | 98.09 | 96.40 | 98.21 | 97.30 | 98.09 |
| | $\Delta_\text{New}$ ↑ | -1.97 | 1.71 | 0.43 | 1.38 | -1.42 | 1.16 | 0.50 | 0.83 | -0.86 | 0.95 | 0.04 | 0.83 |
| | $\mathcal{A}_\text{H}$ | 94.47 | 96.95 | 96.43 | 97.79 | 95.94 | 96.85 | 96.58 | 96.74 | 96.05 | 97.92 | 96.30 | 97.19 |
| **Stanford Cars** | $\mathcal{A}_\text{Base}$ | 78.12 | 78.33 | 70.49 | 69.53 | 75.00 | 74.16 | 72.94 | 74.01 | 78.60 | 78.07 | 78.27 | 77.95 |
| | $\mathcal{A}_\text{New}$ | 60.40 | 69.87 | 73.59 | 78.95 | 63.45 | 72.17 | 74.00 | 75.15 | 73.50 | 76.12 | 74.97 | 75.75 |
| | $\Delta_\text{New}$ ↑ | -14.49 | -5.02 | -1.30 | 4.06 | -11.44 | -2.72 | -0.89 | 0.26 | -1.39 | 1.23 | 0.08 | 0.86 |
| | $\mathcal{A}_\text{H}$ | 68.13 | 73.86 | 72.01 | 73.94 | 68.74 | 73.15 | 73.47 | 74.58 | 75.96 | 77.08 | 76.58 | 76.83 |
| **Flowers102** | $\mathcal{A}_\text{Base}$ | 97.60 | 97.21 | 94.87 | 94.52 | 96.89 | 97.11 | 95.92 | 96.85 | 93.50 | 97.91 | 98.07 | 97.69 |
| | $\mathcal{A}_\text{New}$ | 59.67 | 72.36 | 71.75 | 77.54 | 70.02 | 73.49 | 72.46 | 75.59 | 74.50 | 76.59 | 76.50 | 78.49 |
| | $\Delta_\text{New}$ ↑ | -18.13 | -5.44 | -6.05 | -0.26 | -7.78 | -4.31 | -5.34 | -2.21 | -3.30 | -1.21 | -1.30 | 0.69 |
| | $\mathcal{A}_\text{H}$ | 74.06 | 82.96 | 81.71 | 85.19 | 81.29 | 83.66 | 82.56 | 84.91 | 82.93 | 85.95 | 85.95 | 87.04 |
| **Food101** | $\mathcal{A}_\text{Base}$ | 88.33 | 88.59 | 90.70 | 91.33 | 88.88 | 90.35 | 90.71 | 91.40 | 90.70 | 92.94 | 90.67 | 91.07 |
| | $\mathcal{A}_\text{New}$ | 82.26 | 90.12 | 91.29 | 94.79 | 88.95 | 92.27 | 92.05 | 93.12 | 91.70 | 92.76 | 91.53 | 92.25 |
| | $\Delta_\text{New}$ ↑ | -8.96 | -1.10 | 0.07 | 3.57 | -2.27 | 1.05 | 0.83 | 1.90 | 0.48 | 1.54 | 0.31 | 1.03 |
| | $\mathcal{A}_\text{H}$ | 85.19 | 89.35 | 90.99 | 93.03 | 88.91 | 91.30 | 91.38 | 92.25 | 91.20 | 92.85 | 91.10 | 91.66 |
| **FGVC Aircraft** | $\mathcal{A}_\text{Base}$ | 40.44 | 41.24 | 33.41 | 35.12 | 38.33 | 38.75 | 37.44 | 37.41 | 43.30 | 42.26 | 42.73 | 41.63 |
| | $\mathcal{A}_\text{New}$ | 22.30 | 33.83 | 23.71 | 36.36 | 25.27 | 31.36 | 35.61 | 37.79 | 37.20 | 37.54 | 37.87 | 40.61 |
| | $\Delta_\text{New}$ ↑ | -13.99 | -2.46 | -12.58 | 0.07 | -11.02 | -4.93 | -0.68 | 1.50 | 0.91 | 1.25 | 1.58 | 4.32 |
| | $\mathcal{A}_\text{H}$ | 28.75 | 37.17 | 27.74 | 35.73 | 30.46 | 34.67 | 36.50 | 37.60 | 40.02 | 39.76 | 40.15 | 41.11 |
| **SUN397** | $\mathcal{A}_\text{Base}$ | 80.60 | 80.63 | 79.74 | 80.36 | 80.27 | 79.54 | 80.82 | 81.24 | 81.00 | 83.04 | 82.67 | 82.71 |
| | $\mathcal{A}_\text{New}$ | 65.89 | 72.11 | 76.86 | 78.97 | 74.36 | 76.11 | 78.70 | 82.15 | 79.30 | 79.92 | 78.47 | 79.73 |
| | $\Delta_\text{New}$ ↑ | -9.46 | -3.24 | 1.51 | 3.62 | -0.99 | 0.76 | 3.35 | 6.80 | 3.95 | 4.57 | 3.12 | 4.38 |
| | $\mathcal{A}_\text{H}$ | 72.51 | 76.13 | 78.27 | 79.66 | 77.20 | 77.79 | 79.75 | 81.69 | 80.14 | 81.45 | 80.52 | 81.19 |
| **DTD** | $\mathcal{A}_\text{Base}$ | 79.44 | 79.51 | 77.01 | 75.92 | 77.08 | 76.68 | 80.36 | 80.05 | 80.80 | 80.14 | 83.37 | 84.04 |
| | $\mathcal{A}_\text{New}$ | 41.18 | 54.24 | 56.00 | 59.84 | 53.62 | 59.97 | 59.18 | 63.13 | 58.60 | 63.32 | 62.97 | 63.06 |
| | $\Delta_\text{New}$ ↑ | -18.72 | -5.66 | -3.90 | -0.06 | -6.28 | 0.07 | -0.72 | 3.23 | -1.3 | 3.42 | 3.07 | 3.16 |
| | $\mathcal{A}_\text{H}$ | 54.24 | 64.49 | 64.85 | 66.93 | 63.24 | 67.30 | 68.16 | 70.59 | 67.93 | 70.74 | 71.75 | 72.05 |
| **EuroSAT** | $\mathcal{A}_\text{Base}$ | 92.19 | 91.98 | 87.49 | 81.24 | 91.67 | 90.42 | 94.07 | 94.27 | 97.50 | 92.18 | 92.90 | 93.66 |
| | $\mathcal{A}_\text{New}$ | 54.74 | 78.29 | 60.04 | 66.87 | 58.31 | 67.02 | 73.23 | 75.11 | 64.10 | 71.01 | 73.90 | 77.12 |
| | $\Delta_\text{New}$ ↑ | -9.31 | 14.24 | -4.01 | 2.82 | -5.74 | 2.97 | 9.18 | 11.06 | 0.05 | 6.96 | 9.85 | 13.07 |
| | $\mathcal{A}_\text{H}$ | 68.69 | 84.58 | 71.21 | 73.36 | 71.28 | 76.98 | 82.35 | 83.61 | 77.35 | 80.22 | 82.32 | 84.59 |
| **UCF101** | $\mathcal{A}_\text{Base}$ | 84.69 | 89.15 | 82.33 | 84.86 | 80.07 | 83.37 | 83.00 | 83.43 | 85.70 | 85.95 | 87.10 | 87.79 |
| | $\mathcal{A}_\text{New}$ | 56.05 | 70.16 | 73.45 | 78.23 | 74.50 | 74.77 | 78.66 | 81.40 | 79.30 | 79.44 | 78.80 | 79.95 |
| | $\Delta_\text{New}$ ↑ | -21.45 | -7.34 | -4.05 | 0.73 | -3.00 | -2.73 | 1.16 | 3.90 | 1.80 | 1.94 | 1.30 | 2.45 |
| | $\mathcal{A}_\text{H}$ | 67.46 | 78.52 | 77.64 | 81.41 | 77.18 | 78.84 | 80.77 | 82.40 | 82.38 | 82.57 | 82.74 | 83.69 |

Table 6: **Few-shot cross-dataset generalization** where CLIP is prompt-tuned on the source dataset ImageNet (16 shots per class) and tested on both ImageNet and 10 target datasets. We compare the test set accuracy $\mathcal{A}$ and the accuracy change $\Delta_{\mathcal{A}}$ (higher is better) between pre-trained and prompt-tuned models to quantify generalization and concept forgetting on each target dataset.

| | | | CoOp | | CoCoOp | | PromptSRC | |
|---|---|---|---|---|---|---|---|---|
| | | +Proxy-FDA | ✗ | ✓ | ✗ | ✓ | ✗ | ✓ |
| **Source** | ImageNet | $\mathcal{A}$ | **71.51** | 71.36 | 71.02 | **71.24** | 71.27 | **71.32** |
| | | $\Delta_{\mathcal{A}} \uparrow$ | **4.78** | 4.63 | 4.29 | **4.51** | 4.54 | **4.59** |
| **Target** | **Avg across 10 datasets** | $\mathcal{A}$ | 63.88 | **66.09** | 65.74 | **66.48** | 65.81 | **66.86** |
| | | $\Delta_{\mathcal{A}} \uparrow$ | -1.20 | **1.01** | 0.66 | **1.40** | 0.72 | **1.78** |
| | Caltech101 | $\mathcal{A}$ | 93.70 | 94.35 | 94.43 | 94.51 | 93.60 | 94.42 |
| | | $\Delta_{\mathcal{A}} \uparrow$ | 0.76 | 1.41 | 1.49 | 1.57 | 0.66 | 1.48 |
| | OxfordPets | $\mathcal{A}$ | 89.14 | 90.53 | 90.14 | 90.62 | 90.25 | 90.78 |
| | | $\Delta_{\mathcal{A}} \uparrow$ | -0.07 | 1.32 | 0.93 | 1.41 | 1.04 | 1.57 |
| | Stanford Cars | $\mathcal{A}$ | 64.51 | 66.18 | 65.32 | 66.22 | 65.70 | 66.55 |
| | | $\Delta_{\mathcal{A}} \uparrow$ | -0.81 | 0.86 | 0.00 | 0.90 | 0.38 | 1.23 |
| | Flowers102 | $\mathcal{A}$ | 68.71 | 71.54 | 71.88 | 72.32 | 70.25 | 72.04 |
| | | $\Delta_{\mathcal{A}} \uparrow$ | -2.63 | 0.20 | 0.54 | 0.98 | -1.09 | 0.70 |
| | Food101 | $\mathcal{A}$ | 85.30 | 86.86 | 86.06 | 86.91 | 86.15 | 87.38 |
| | | $\Delta_{\mathcal{A}} \uparrow$ | -0.76 | 0.80 | 0.00 | 0.85 | 0.09 | 1.32 |
| | FGVC Aircraft | $\mathcal{A}$ | 18.47 | 22.09 | 22.94 | 23.49 | 23.90 | 24.79 |
| | | $\Delta_{\mathcal{A}} \uparrow$ | -6.25 | -2.63 | -1.78 | -1.23 | -0.82 | 0.07 |
| | SUN397 | $\mathcal{A}$ | 64.15 | 66.12 | 67.36 | 67.62 | 67.10 | 67.53 |
| | | $\Delta_{\mathcal{A}} \uparrow$ | 1.65 | 3.62 | 4.86 | 5.12 | 4.60 | 5.03 |
| | DTD | $\mathcal{A}$ | 41.92 | 45.13 | 45.73 | 46.15 | 46.87 | 47.31 |
| | | $\Delta_{\mathcal{A}} \uparrow$ | -2.47 | 0.74 | 1.34 | 1.76 | 2.48 | 2.92 |
| | EuroSAT | $\mathcal{A}$ | 46.39 | 49.08 | 45.37 | 47.89 | 45.50 | 48.37 |
| | | $\Delta_{\mathcal{A}} \uparrow$ | -1.21 | 1.48 | -2.23 | 0.29 | -2.10 | 0.77 |
| | UCF101 | $\mathcal{A}$ | 66.55 | 69.01 | 68.21 | 69.10 | 68.75 | 69.42 |
| | | $\Delta_{\mathcal{A}} \uparrow$ | -0.20 | 2.26 | 1.46 | 2.35 | 2.00 | 2.67 |

Table 7: **Few-shot prompt tuning in both base-to-new class generalization and domain generalization settings.** Here we compare with more recent prompt tuning methods. Note both OGEN and our Proxy-FDA are plugged into the PromptSRC baseline. For fair comparison with CLAP, we obtain its base-to-new generalization results by re-running its official codes with the ViT-B/16 backbone used by all other methods. The domain generalization results of CLAP are directly extracted from the CLAP paper. $\mathcal{A}_{\text{H}}$ denotes the Harmonic mean of $\mathcal{A}_{\text{Base}}$ and $\mathcal{A}_{\text{New}}$.

| | | Base-to-New Class Generalization | | | | Domain Generalization | | | | |
|---|---|---|---|---|---|---|---|---|---|---|
| | | Avg across 11 datasets | | | | $\mathcal{A}_{\text{Source}}$ | $\mathcal{A}_{\text{Target}}$ | | | |
| | | $\mathcal{A}_{\text{Base}}$ | $\mathcal{A}_{\text{New}}$ | $\Delta_{\text{New}} \uparrow$ | $\mathcal{A}_{\text{H}}$ | ImageNet | -V2 | -Sketch | -A | -R |
| **Text Knowledge from LLM** | ProText | 72.95 | 76.98 | 2.76 | 74.91 | 70.22 | 63.54 | 49.45 | 51.47 | 77.35 |
| | ArGue-N | 83.77 | **78.74** | **4.52** | **81.18** | 71.84 | 65.02 | 49.25 | 51.47 | 76.96 |
| **Regularization method** | OGEN | 84.17 | 76.86 | 2.64 | 80.34 | 73.13 | 65.37 | 48.96 | 50.75 | 77.12 |
| | CLAP | 84.34 | 76.62 | 2.40 | 80.29 | 73.38 | 65.00 | 48.35 | 49.53 | 77.26 |
| | Proxy-FDA | **84.47** | 77.45 | 3.23 | 80.81 | **73.44** | **65.79** | **49.83** | **51.54** | **77.45** |

Table 8: **Continual fine-tuning: test accuracy $\mathcal{A}_{\mathbf{LP}}$ and $\Delta_{\mathbf{LP}}$ for models fine-tuned on three task sequences**. The first 3 rows show performance on fine-tuned tasks and the 4th row shows performance averaged on 6 other datasets, comparing our method with 5 classic continual learning methods.

| Fine-tune dataset | Evaluation dataset | LwF $\mathcal{A}_{LP}$ | $\Delta_{LP}\uparrow$ | LFL $\mathcal{A}_{LP}$ | $\Delta_{LP}\uparrow$ | iCaRL $\mathcal{A}_{LP}$ | $\Delta_{LP}\uparrow$ | D+R $\mathcal{A}_{LP}$ | $\Delta_{LP}\uparrow$ | ZSCL $\mathcal{A}_{LP}$ | $\Delta_{LP}\uparrow$ | FDA (ours) $\mathcal{A}_{LP}$ | $\Delta_{LP}\uparrow$ | Proxy-FDA (ours) $\mathcal{A}_{LP}$ | $\Delta_{LP}\uparrow$ |
|---|---|---|---|---|---|---|---|---|---|---|---|---|---|---|---|
| SVHN→CIFAR10→RESISC45 | SVHN | 90.48 | -3.81 | 91.90 | -3.21 | 91.62 | -3.67 | 93.30 | -2.78 | 92.70 | -3.23 | **96.77** | 0.61 | 96.72 | **0.93** |
| | CIFAR10 | 93.90 | -2.90 | 94.88 | -2.32 | 95.17 | -2.10 | 95.41 | -1.90 | 95.82 | -1.60 | 97.13 | 0.57 | **97.29** | **1.02** |
| | RESISC45 | 94.22 | 3.10 | 93.90 | 2.98 | 93.72 | 2.83 | 94.94 | 3.68 | 94.89 | 3.62 | 95.22 | 4.14 | **95.38** | **4.22** |
| | Others | 80.73 | -4.20 | 81.31 | -3.76 | 80.78 | -4.11 | 81.86 | -3.20 | 83.10 | -2.80 | **87.21** | 0.76 | 86.95 | **1.08** |
| SVHN→CIFAR100→RESISC45 | SVHN | 89.48 | -4.34 | 90.29 | -4.08 | 90.97 | -4.31 | 92.30 | -3.23 | 91.81 | -3.92 | 96.18 | 0.63 | **96.43** | **0.71** |
| | CIFAR100 | 83.24 | -3.25 | 83.95 | -3.01 | 84.06 | -3.13 | 84.82 | -2.60 | 85.07 | -2.13 | **86.33** | 0.72 | 86.14 | **0.85** |
| | RESISC45 | 93.80 | 3.21 | 94.91 | 3.62 | 94.87 | 3.54 | 95.08 | 3.71 | 94.96 | 3.65 | 95.32 | 3.95 | **95.46** | **4.01** |
| | Others | 81.73 | -4.11 | 82.04 | -3.80 | 81.62 | -4.02 | 82.17 | -3.43 | 82.86 | -3.11 | 89.02 | 0.68 | **89.09** | **0.96** |
| SVHN→Cars→RESISC45 | SVHN | 91.43 | -3.64 | 92.74 | -2.92 | 91.75 | -3.13 | 92.86 | -2.84 | 92.98 | -2.72 | 96.74 | 0.79 | **96.91** | **0.94** |
| | Cars | 81.69 | -2.79 | 81.82 | -2.64 | 81.70 | -2.80 | 82.11 | -2.12 | 82.68 | -1.84 | **84.38** | 1.14 | 84.32 | **1.36** |
| | RESISC45 | 93.92 | 3.34 | 94.96 | 3.55 | 94.97 | 3.58 | 95.19 | 3.72 | 95.04 | 3.63 | 95.12 | 3.92 | **95.23** | **4.07** |
| | Others | 81.63 | -4.07 | 82.24 | -3.60 | 81.88 | -3.89 | 82.73 | -3.12 | 83.10 | -2.80 | 89.54 | 0.96 | **89.67** | **1.17** |

Table 9: **Continual fine-tuning:** comparing the average accuracy on Split ImageNet-R.

| L2P | DualPrompt | CODA-Prompt | Continual-CLIP | SLCA | LDIFS | FDA (ours) | Proxy-FDA (ours) |
|---|---|---|---|---|---|---|---|
| 74.60±1.21 | 77.24±1.27 | 78.13±1.18 | 76.23±1.18 | 81.22±1.23 | 83.62±1.16 | 85.97±1.05 | **86.71±1.24** |

Table 10: **Knowledge distillation:** comparing the top-1 accuracy on ImageNet.

| Teacher | Student | Logits-based KD++ | DIST | WTTM | Feature-based CNA | ITRD | FDA (ours) | Proxy-FDA (ours) |
|---|---|---|---|---|---|---|---|---|
| ResNet-34 (73.31) | ResNet-18 (69.76) | 71.98 | 72.07 | **72.19** | 71.38 | 71.68 | 72.02 | 72.17 |
| ResNet-50 (76.16) | MobileNet (68.87) | 72.77 | 73.24 | 73.09 | 72.39 | - | 73.31 | **73.45** |

