# OpenReview forum: "Proxy-FDA: Proxy-based Feature Distribution Alignment for Fine-tuning Vision Foundation Models without Forgetting"
_ICLR.cc/2025/Conference — Submitted to ICLR 2025_

### Official Review · Reviewer_DVQH · 2024-11-01

**Soundness:** 3
**Presentation:** 3
**Contribution:** 3
**Rating:** 6
**Confidence:** 4

**Summary:**

This paper introduces Proxy-FDA, a structural regularization method designed to mitigate concept forgetting in fine-tuning large pre-trained vision models. Proxy-FDA achieves this by preserving local feature neighborhood structures in the pre-trained feature space through
Feature Distribution Alignment (FDA), leveraging nearest neighbor graphs. Additionally, Proxy-FDA employs dynamic proxy generation to increase data diversity, which enhances feature alignment. The method is evaluated in few-shot and continual fine-tuning contexts,
demonstrating strong performance.

**Strengths:**

1. Preserving the neighbor structure within the pre-trained feature space beyond class semantics is novel and may bring some insight to the model adaptation community.
2. State-of-the-art performance validates the effectiveness of the proposed approach in reducing concept forgetting.
3. This paper is well-written and organized.

**Weaknesses:**

1. As the output pooling weights are softmax-normalized and convex combinations are expected, figure 2(c) may be problematic. Neither the synthesized proxy P+ nor P- lies within the convex hull of corresponding neighbors.
2. In L261, why do both the positive set and the negative set lack diversity? How does hard-class mining lead to the semi-hard nature of X^-? And how does this nature alleviate “limited diversity”? More explanations for the above questions should be provided.

**Questions:**

1. How is the number of proxy positives n^{p+} and negatives n^{p-} determined? Is it dynamic across different batches?
2. How can the generated proxies within the convex hull help shape the decision boundary?

---

> ### Author Response · Authors · 2024-11-23
>
> Thank you for the recognition of our work and constructive feedback. We spent quite some time updating the paper to include the requested experiments and clarifications by all the reviewers. We respond to your specific comments below.
>
> **In Fig. 2(c), proxies do not lie in the convex hull of neighbors. How proxies in the convex hull shape the decision boundary?**
>
> Thanks for pointing this out. Although Fig. 2(c) is only for illustration purpose, we will modify it to reflect that the generated proxies lie in the convex hull of the corresponding set of positive features ($\boldsymbol{X}\_{i}^{+}$) or negative features ($\boldsymbol{X}\_{i}^{-}$).
>
> Regarding the other question, we conjecture that proxies generated in a convex hull can be key to reshape the decision boundary, especially when there is overlap between the positive and negative feature manifolds — Fig. 1(b) shows how typical such overlap is in the feature space. In case of feature manifold overlap, sparsely sampled feature points won't be able to characterize a fine-grained decision boundary, while we densify feature distributions using proxies to refine that boundary. Note from the performance perspective, it's not enough to generate proxies only around the decision boundary (Fig. 6 and L949-953 show sub-optimal performance). This indicates the key role of proxy diversity in enhancing FDA with both richer data and fine-grained decision boundary.
>
> **More explanations of limited data diversity and how hard-class mining alleviates limited diversity.**
>
> Due to the batch size constraint, the sampled data in $\boldsymbol{X}\_{i}^{+}$ and $\boldsymbol{X}\_{i}^{-}$ are intrinsically limited in both quantity and diversity. It's more so in few-shot learning tasks where only a couple of training instances can be sampled from each class.
>
> To alleviate the data challenge, we perform hard-class mining within batch (besides generating proxies). Appendix A details the hard mining algorithm that greedily samples related class data. This means $\boldsymbol{X}\_{i}^{-}$ includes negative features that are close to the *anchor* $\boldsymbol{x}\_{i}$, but still further away from $\boldsymbol{x}\_{i}$ than the positive features $\boldsymbol{X}\_{i}^{+}$. The metric learning community calls such $\boldsymbol{X}\_{i}^{-}$ as semi-hard negatives. They often provide an informative training signal (i.e., relatively large loss), without sampling the vast space of *easy* negatives with near-zero loss values. Hence, semi-hard negatives alleviate the limited data issue. L260-262 provides such rationale, but we will make it clearer.
>
> **How to determine the number of positive proxies $n^{p+}$ and negative proxies $n^{p-}$?**
>
> Appendix D (L862-863) sets $n^{p+}$ and $n^{p-}$ proportional to the number of sampled positives $\boldsymbol{X}\_{i}^{+}$ and negatives $\boldsymbol{X}\_{i}^{-}$, respectively, using a scalar $s$ which is fixed across batches. Fig. 5(d) shows the sensitivity analysis for $s$.

---

> > ### Comment · Reviewer_DVQH · 2024-11-26
> >
> > My questions are well addressed. I will keep my original rating.

---

> > > ### Author Response · Authors · 2024-12-01
> > >
> > > Thanks for your feedback. We will continue to integrate the new insights/results into the paper.

---

### Official Review · Reviewer_VrYK · 2024-11-03

**Soundness:** 3
**Presentation:** 3
**Contribution:** 2
**Rating:** 6
**Confidence:** 4

**Summary:**

This paper presents a new approach to mitigate concept forgetting in model fine-tuning (robust fine-tuning) by building on existing feature-matching methods. Specifically, the authors aim to align the feature structure by regularizing the feature space using k-nearest neighbors (KNN) within each batch. They also propose generating proxies from the data to preserve diversity across datasets.

**Strengths:**

- Robust fine-tuning is a highly active and valuable area of research with significant relevance and potential.
- The motivation for the proposed method is strong, as preserving data structure during feature matching is both reasonable and innovative.
- The writing is clear, and the approach is presented in a way that is easy to follow.

**Weaknesses:**

- Motivation and Selection of Distribution Alignment Method: The motivation for using distribution alignment through feature matching, specifically with regularization on nearest-neighbor features, is not fully explained. This approach resembles knowledge distillation; therefore, clarification on how this method differs from distillation between the original and fine-tuned models would strengthen the argument.

- Design of Equation 2: Equation 2 appears to follow a sigmoid loss structure but replaces traditional labels (+1, -1) with a weight, $w_{ij}$​. Is this weight defined as $w_{ij}=\cos⁡(x_i,x_j)$? An additional explanation of the rationale and comparative benefits of this design choice would be helpful.

- Clustering with KNN: The method clusters features in each batch using KNN, which may not adequately group samples with the same or similar labels, especially during fine-tuning. The clustering might benefit from label-based constraints, either including or excluding samples based on label proximity, to enhance feature alignment.

- Proxy Generator Motivation: The purpose and function of the proxy generator remain unclear. Specifically, the reason for incorporating an attention layer is not well-justified, as it is unclear how it assists in integrating information across positive and negative feature sets. An ablation study or additional analysis of the generator architecture would clarify its contribution and effectiveness.

- (minor) Some descriptions need clarification:
 Line 167: What is meant by "low task loss $L_{task}$​"?
 Line 168: Please clarify "whilst preventing concept forgetting on any target dataset $D\neq D_{ft}$".

**Questions:**

See the weakness above.
Generally, this is a solid paper. However, some remaining concerns should be further discussed. I may adjust the score according to the response from the authors.

---

> ### Author Response · Authors · 2024-11-23
> **Author rebuttal (Part 1)**
>
> Thanks for the recognition of our work and helpful suggestions for improvement. We spent quite some time integrating the requested clarifications into the paper, and the paper is also updated to include supporting ablation studies. Below is our point-by-point response.
>
> **Motivation of using kNN feature-based FDA, and differences from knowledge distillation.**
>
> As a quick recap, our introduction section (L46-85) motivates that for forgetting mitigation, feature regularization is often better than weight-space regularization, but existing feature regularization methods are point-wise and lack explicit awareness of feature neighborhood structures (hence still sub-optimal). Then for the first time in forgetting mitigation, we propose a structure-wise feature regularization method Proxy-FDA, which preserves structural knowledge by matching kNN feature graphs. As mentioned in L146-152, Proxy-FDA is indeed similar to Knowledge Distillation (KD). Proxy-FDA can be seen as a relational KD method in general, but differs from prior works in distilling knowledge from both neighbor indices and similarities, with an additional proxy learning component. To show the utility of Proxy-FDA in the KD field, we have conducted some KD experiments on ImageNet after paper submission deadline. Please refer to Table 10 and L1005-1018 in the updated paper. Results show the competitive performance of Proxy-FDA and its advantages over related KD baselines.
>
> **Explain and compare the weighting mechanism in Eq. (2).**
>
> As stated in L209, Eq. (2) defines a re-weighted Sigmoid loss where the weighting is derived by the cosine similarity $\hat{w}\_{ij} = \cos(\hat{\boldsymbol{x}}\_{i},\hat{\boldsymbol{x}}\_{j})$ between pre-trained features $(\hat{\boldsymbol{x}}\_{i},\hat{\boldsymbol{x}}\_{j})$, not between features $(\boldsymbol{x}\_{i},\boldsymbol{x}\_{j})$ being fine-tuned. The goal is to transfer the nearest neighbor graph (with both neighbor indices and similarities) from the pre-trained model to fine-tuned model. We conjecture that the structural knowledge contained in a complete nearest neighbor graph is richer than the knowledge in neighbor indices only. Appendix E (L912-914) and Fig. 6 show that we will see notable performance drop if we only transfer neighbor indices (i.e., setting $\hat{w}\_{ij}=1$).

---

> > ### Author Response · Authors · 2024-11-23
> > **Author rebuttal (Part 2)**
> >
> > **Will clustering informed of label similarity improve FDA?**
> >
> > Great question! In response, we use class labels for task learning with $\mathcal{L}\_{\text{task}}$, while $\mathcal{L}\_{\text{FDA}}$ is treated as a regularization term without involving labels. The intuition behind the label-free FDA loss is that we aim to preserve a foundation model's general knowledge that is often richer than class labels on downstream datasets. Particularly, we design $\mathcal{L}\_{\text{FDA}}$ to preserve the structural knowledge among nearest neighbor graphs, and the graphs only involve feature (not label) similarities in a large enough feature neighborhood $R\_i$ (size $K>n$). Fig. 1(b) shows such structural knowledge can go beyond class concepts in $R_i$ (e.g., cross-class attributes), which is important to maintain the generalizability of foundation model. On the other hand, introducing label information from downstream tasks into the construction or modeling of neighbor graphs may end up aligning class semantics during FDA, thus may risk forgetting the desired general knowledge.
> >
> > To provide empirical support, we compare with an FDA variant that constructs and models nearest neighbor graphs using both feature similarities $\hat{w}\_{ij}$ and label similarities $w\_{ij}^t$. Note we use the text encoder of CLIP to compute $w\_{ij}^t$ as the text-text similarity between the prompt templates "a photo of a {class}" of different classes. The table below shows the comparison results for few-shot prompt tuning in the base-to-new setting (average across 11 datasets). Results confirm that the use of $w\_{ij}^t$ leads to much lower $\mathcal{A}\_{\text{New}}$, i.e., worse generalization on unseen classes.
> > |                                              | $\mathcal{A}\_{\text{Base}}$ | $\mathcal{A}\_{\text{New}}$ | $\mathcal{A}\_{\text{H}}$ |   |
> > |----------------------------------------------|------------------------------|-----------------------------|---------------------------|---|
> > | CoOp                                         |                        82.69 |                       63.22 |                     71.66 |   |
> > | +Proxy-FDA ($\hat{w}\_{ij}$ - default)       |                    **83.16** |                   **73.67** |                 **78.13** |   |
> > | +Proxy-FDA ($\hat{w}\_{ij} \cdot w\_{ij}^t$) |                        83.02 |                       70.91 |                     76.49 |   |
> > | PromptSRC                                    |                        84.26 |                        76.1 |                     79.97 |   |
> > | +Proxy-FDA ($\hat{w}\_{ij}$ - default)       |                        84.47 |                   **77.45** |                 **80.81** |   |
> > | +Proxy-FDA ($\hat{w}\_{ij} \cdot w\_{ij}^t$) |                    **84.55** |                       77.12 |                     80.66 |   |
> >
> > **Analysis and ablation on the attention layer used in proxy generator.**
> >
> > Sorry about the missing details. We learn the proxy generator to produce unseen data out of diverse feature combinations within the positive set $\boldsymbol{X}\_{i}^{+}$ or negative set $\boldsymbol{X}\_{i}^{-}$. The attention layer helps to achieve this goal by first modeling the global context among all input features with pairwise attention — attention mask is used to ensure the independence between $\boldsymbol{X}\_{i}^{+}$ and $\boldsymbol{X}\_{i}^{-}$ as mentioned in L773. Convolutional layers, however, have only local receptive fields and have to rely on pooling operations to capture long-range dependencies. We have tried replacing the attention layer with convolutional plus pooling layers, and making sure the resulting proxy generator has a similar parameter count. Fig. 6 in the updated paper shows the attention-free proxy generator achieves consistently lower performance. We also updated Appendix B (L772) and E (L916-944) to include the reasoning.
> >
> > **Clarifying arguments in L167-168.**
> >
> > In Line 167, "low task loss" means the downstream task loss $\mathcal{L}\_{\text{task}}$ is minimized, i.e., good fine-tuning performance is achieved. In Line 168, "whilst preventing concept forgetting on any target dataset $\mathcal{D} \neq \mathcal{D}\_{\text{ft}}$" means that after fine-tuning on $\mathcal{D}\_{\text{ft}}$, we aim for good generalization with no forgetting on any unseen dataset $\mathcal{D}$ that is different from $\mathcal{D}\_{\text{ft}}$.

---

> > > ### Author Response · Authors · 2024-12-01
> > > **Gentle reminder of the discussion deadline**
> > >
> > > Dear Reviewer VrYK, thanks again for your insightful feedback! We are sending you this gentle reminder since the rebuttal deadline is approaching. Would you mind letting us know if your concerns have been fully addressed or if you have more questions? In the latter case, we will be happy to follow up. Looking forward to your reply and thanks for your time!

---

### Official Review · Reviewer_B3Zd · 2024-11-03

**Soundness:** 2
**Presentation:** 2
**Contribution:** 2
**Rating:** 6
**Confidence:** 3

**Summary:**

This paper introduces Proxy-FDA, a feature-based regularization method for fine-tuning vision foundation models (e.g., CLIP, DINOv2) on downstream tasks. Unlike point-wise regularization, FDA preserves the structure-wise knowledge by aligning the feature space of the pre-trained and fine-tuned models, incorporating local neighborhood structures. Proxy-FDA further generates synthetic features (proxies) to increase data diversity. The method is evaluated across end-to-end fine-tuning, few-shot tuning, and continual learning scenarios.

**Strengths:**

-  The method is well-motivated.
- The evaluation seems comprehensive and the results appear impressive.

**Weaknesses:**

- Some recent related works lack discussion in the related work section but are directly compared in the experimental section, such as CLIPood and PromptSRC, which are both regularization-based fine-tuning approaches. Including a discussion of these methods in the related work section would provide better context for readers.
- Several technical details require further clarification. (1) Details about the network architecture and training parameters for the proxy generator are absent. (2) The hard class mining strategy specifies $n=4$ (Line 777), but it is unclear how this approach is used in few-shot cases (e.g., 1- or 2-shot) (3) The scalar $s=0.4$ is not introduced or explained in Section 3.2 but it is discussed and analyzed in hyper-parameter analysis (e.g., Figure 5)
- Important implementation details are not provided. (1) Detailed specifications for hype-parameters (e.g., temperature $\tau$, bias $b$, loss coefficient $\lambda$) are missing. (2) The specific $K$ value for each dataset is also absent. Providing these details would benefit the community. (3) Implementation details are missing for the continual fine-tuning setting (e.g., Table 8), making it difficult to confirm evaluation fairness across methods.
- Evaluation should be improved. (1) How does FDA/Proxy-FDA perform when compared to previous SoTA FD-Align in its official evaluation setting (e.g., compare APE-T+FD-Align and APE-T+Proxy-FDA by training the model on ImageNet and testing on ImageNet V2/Sketch) ? (2) CLAP [1], another regularization-based linear probing method, seems more efficient as it does not require a validation set for extensive hyper-parameter selection. Including a comparison with CLAP would clarify the strengths and limitations of the proposed method.
- Ablation study on batch size. Since the FDA relies on batch-based feature alignment, it is essential to analyze the impact of varying batch sizes.

[1] A Closer Look at the Few-Shot Adaptation of Large Vision-Language Models. In CVPR, 2024

**Questions:**

- Please find the weaknesses.

---

> ### Author Response · Authors · 2024-11-22
>
> Thanks for the detailed feedback! We spent quite some time on the requested comparisons and ablations, as well as paper revision to include the new results, analysis and clarifications. We respond to specific comments below.
>
> **Discuss CLIPood and PromptSRC in the related work section.**
>
> Thanks for the reminder. The discussion is added in L135-137.
>
> **Clarifications on some technical details.**
>
> Sorry about the missing details. 1) Appendix B details about the proxy generator architecture, including one attention layer with attention mask applied, and two convolutional layers for proxy pooling. Please see text for the feature dimensionality, output channel size and normalization. Appendix D (L898-904) is updated to elaborate on the parameter count and compute cost. We plan to release codes in the future to provide more details. 2) In few-shot settings, when fewer than $n=4$ samples are available for each class, we perform random data augmentation to guarantee 4 samples per class (clarified in L856-858). In this case, FDA aligns the structural similarity between the original and augmented data. 3) We updated Section 3.2 to introduce the scalar $s$ (L266), which is then referenced in Appendix D.
>
> **Missing hype-parameters, per-dataset $K$, and implementation details for continual learning.**
>
> In response: 1) both temperature $\tau$ and bias $b$ are learnable parameters that are initialized as similarly in (Zhai et al., 2023). Such initialization helps to handle the imbalance between positives and negatives in the loss function. We updated L210-211 to reflect this. While $\lambda$ is the loss coefficient tuned on the validation set of each task, as mentioned in L895. 2) Fig. 5 is updated to include the optimal $K$ for each dataset. 3) We also update Appendix F (L994-1001) to include the dataset splitting and learning details on Split ImageNet-R. The same implementation details enable fair comparisons with other methods.
>
> **Comparison with FD-Align and CLAP.**
>
> As suggested, we compare Proxy-FDA with FD-Align in the domain generalization setting, i.e., prompt tuning on ImageNet (16 shots per class) before evaluating OOD generalization on ImageNetV2 and ImageNet-Sketch. We use the same ViT-B/32 backbone for fair comparison. The table below confirms our advantage in this setting.
> |                                                  | APE-T | APE-T+FD-Align | APE-T+Proxy-FDA |   |
> |--------------------------------------------------|-------|----------------|-----------------|---|
> | ImageNet ($\mathcal{A}$)                         | 68.74 |          69.15 | **69.48**  |   |
> | ImageNet ($\Delta_{\mathcal{A}}\uparrow$)        |   5.4 |           5.81 | **6.14**   |   |
> | ImageNetV2 ($\mathcal{A}$)                       | 59.58 |          60.83 | **61.21**  |   |
> | ImageNetV2 ($\Delta_{\mathcal{A}}\uparrow$)      |  3.66 |           4.91 | **5.29**   |   |
> | ImageNet-Sketch ($\mathcal{A}$)                  | 43.23 |          44.04 | **45.94**  |   |
> | ImageNet-Sketch ($\Delta_{\mathcal{A}}\uparrow$) |  0.92 |           1.73 | **3.63**   |   |
>
> Table 7 in the updated paper compares Proxy-FDA with a couple of recent prompt tuning methods, including the regularization method CLAP. Results show our consistent gains over CLAP in the base-to-new generalization setting and the additional domain generalization setting (see details in L975-989). On the other hand, we acknowledge that CLAP shines in that it doesn't use a validation set which is required by other methods (including ours) for hyper-parameter tuning. One of our future plans is to integrate the high-level idea of CLAP (using the Augmented Lagrangian Multiplier approach) into Proxy-FDA, in order to experiment under the validation-free scenario too.
>
> **Ablation on batch size.**
>
> Please refer to Fig. 5(a) in the updated paper for ablation results, and to L851-854 for analysis.

---

> > ### Comment · Reviewer_B3Zd · 2024-11-27
> >
> > Thanks for the authors' response. Some of my concerns have been addressed. I have a remaining key concern regarding the fairness of the comparison with other baselines in the 1-/2- shot few-shot setting if the data augmentation is used to ensure 4 samples per class.

---

> > > ### Author Response · Authors · 2024-12-01
> > > **Re: Official Comment by Reviewer B3Zd**
> > >
> > > Thanks for your feedback. Regarding the data augmentation used for 1- or 2-shot prompt tuning experiments, we found augmentation does not impact performance much, so we didn't mention it as an important implementation detail. Our main purpose of using data augmentation is to fix $n=4$ and keep the same batch size $B=m\cdot n$ across the varying shot learning tasks, without tuning these hyper-parameters in few-shot as well as other types of experiments.
> > >
> > > Below we show our experimental logs that compare the results of our (Proxy-)FDA method with and without using data augmentation in 1- and 2-shot scenarios. Note in these scenarios, no augmentation means $n$ is simply set to 1 or 2. To further ensure fair comparison with the augmentation-free PromptSRC baseline, the previous top performer in Table 2 and Fig 4, we choose $m$ such that we have the same batch size with PromptSRC.
> > >
> > > |                                       |                                 |   1-shot                       |                           |                                 |   2-shot                       |                           |
> > > |---------------------------------------|---------------------------------|--------------------------------|---------------------------|---------------------------------|--------------------------------|---------------------------|
> > > |                                       |   $\mathcal{A}\_{\text{Base}}$  |   $\mathcal{A}\_{\text{New}}$  | $\mathcal{A}\_{\text{H}}$ |   $\mathcal{A}\_{\text{Base}}$  |   $\mathcal{A}\_{\text{New}}$  | $\mathcal{A}\_{\text{H}}$ |
> > > |   PromptSRC+Proxy-FDA (default aug.)  |   75.41                         |   73.60                        | 74.49                     |   77.72                         |   75.03                        | 76.35                     |
> > > |   PromptSRC+FDA (default aug.)        |   75.36                         |   71.34                        | 73.29                     |   77.83                         |   73.18                        | 75.43                     |
> > > |   PromptSRC+Proxy-FDA (w/o aug.)      |   75.45                         |   73.52                        | 74.47                     |   77.74                         |   75.14                        | 76.42                     |
> > > |   PromptSRC+FDA (w/o aug.)            |   75.40                         |   71.22                        | 73.25                     |   77.85                         |   73.09                        | 75.39                     |
> > > |   PromptSRC                           |   75.12                         |   66.26                        | 70.41                     |   77.35                         |   69.24                        | 73.07                     |
> > >
> > > We see that 1) both Proxy-FDA and FDA without using augmentation are on-par with their augmentation-based version. We conjecture that this is because our batch is comprised of similar class data that are sampled via hard class mining. Hence even without data augmentation for the 1 or 2 shots available for each class, we can still have a meaningful modeling and alignment of feature neighborhood structures within batch. 2) The augmentation-free variants still consistently outperform PromptFRC (and all other methods in Table 2/Fig 4), 3) Proxy-FDA is relatively more resistant to the absence of data augmentation than FDA, given the smaller change (or increase) of generalization performance in $\mathcal{A}\_{\text{New}}$. We attribute this to the help of "feature augmentation" in Proxy-FDA.
> > >
> > > We will clarify the above findings in the paper. Please let us know if you have more questions.

---

> > > > ### Comment · Reviewer_B3Zd · 2024-12-01
> > > >
> > > > Thank you for your response, which has addressed my major concerns, and I have decided to raise my score to 6.

---

> > > > > ### Author Response · Authors · 2024-12-01
> > > > > **Re: Official Comment by Reviewer B3Zd**
> > > > >
> > > > > Thanks for raising the score! We will continue to integrate the new insights/results into the paper.

---

### Official Review · Reviewer_5Vid · 2024-11-04

**Soundness:** 3
**Presentation:** 3
**Contribution:** 2
**Rating:** 5
**Confidence:** 4

**Summary:**

This work proposes a regularization method to mitigate the forgetting problem during fine-tuning foundation models by aligning the nearest neighbor graphs between the pre-trained and fine-tuned feature spaces. The authors experimentally demonstrate that this method outperforms baseline models on both few-shot image classification and continual learning tasks.

**Strengths:**

1.This work is clearly presented and easy to understand.

2.Experimental results show that this method outperforms some baseline methods.

**Weaknesses:**

1.why does aligning feature distributions help alleviate forgetting? The essence of feature subspace alignment is to make the feature distributions learned by the pre-trained and fine-tuned models more consistent. In the extreme case, this alignment could lead to a collapse back to the original state.

2.While the authors argue for the advantages of Proxy-FDA, it would be beneficial to include a theoretical foundation or empirical evidence that explains why this approach is expected to outperform traditional methods.

3.One potential concern is that the comparative methods presented by the authors do not include recent works from 2024 onward, which could result in an incomplete or potentially inaccurate assessment of the method's performance.

4.The author’s motivation is difficult to discern from Figure 1, which could benefit from clearer visual cues or annotations to enhance interpretability. Figure 2 could be improved to more effectively present the technology in a clear and understandable manner.

**Questions:**

See weaknesses.

---

> ### Author Response · Authors · 2024-11-22
>
> Thank you for the constructive feedback on our work. We spent quite some time on the requested experiments, and the paper is updated to include changes as suggested by all the reviewers. Below is our point-by-point response to your questions.
>
> **Why does FDA alleviate forgetting? It could collapse back to the original state.**
>
> In response, there are two categories of regularization methods to mitigate forgetting in the literature. Weight-space regularization minimizes the change in model weights that implicitly store the learned knowledge; hence such regularization helps transfer knowledge from a pre-trained model to fine-tuned model (i.e., less forgetting or even positive forward transfer). Another type of regularization happens in the feature space that characterizes knowledge of a model in its input-output behaviour. Feature-space regularization is often found to preserve more knowledge due to the explicit constraint on features (i.e., model behaviour).
>
> In practice, either the weight- or feature-space regularization term is added to the task loss with proper scaling. This can keep the fine-tuned model in the desired vicinity that preserves the pre-trained knowledge while still learning the task at the same time (hence moving away from the pre-trained model). Our (Proxy-)FDA belongs to the feature-space regularization methods, but relaxes the feature constraint from point-wise L2 distance to a structural one. Fig. 3 shows the unique advantage of our structural regularization method over point-wise method LDIFS: we have larger L2 distance (i.e., learning task with a farther away model checkpoint) during FDA, but preserves more knowledge and eventually achieves positive forward transfer. Table 1 confirms that (Proxy-)FDA leads to strong fine-tuning performance as well as significantly reduced forgetting.
>
> **Include theoretical or empirical evidence to explain why Proxy-FDA outperforms traditional methods.**
>
> As a quick recap, we have provided two empirical observations in the paper to justify the advantage of our structure-wise feature regularization method. First, compared to point-wise regularization methods, Proxy-FDA can better preserve the structural knowledge among nearest neighbor graphs in local feature neighborhoods. Fig. 1(b) shows through t-SNE visualization that the structural knowledge is rich enough to go beyond class concepts (e.g., cross-class attributes). Preserving such common-sense knowledge is useful to maintain the generalizability of foundation model, leading to much less forgetting than point-wise regularization methods. Second, Fig. 3 shows a strong correlation between forgetting and a structure-aware distributional distance metric OTDD. As mentioned in L397-398, the strong correlation suggests that having some form of structure-wise feature regularization can mitigate forgetting better than point-wise methods, and our structural method Proxy-FDA is one such instantiation. In other words, the second observation explains our advantage from an optimization perspective. We leave theoretical analysis on how Proxy-FDA improves the fine-tuning generalization error bound for future work.
>
> **Compare with recent works from 2024 onward.**
>
> Thanks for the reminder. In the submitted paper, we treat LDIFS (published in 2024) as the main competitor for end-to-end fine-tuning (Table 1,4) and continual fine-tuning (Table 3,9) experiments. Now the paper is updated to include 1) comparisons with four 2024 works for few-shot prompt tuning. Table 7 and Appendix F (L975-989) include the experimental details and illustrate our benefits. 2) Knowledge distillation experiments where our Proxy-FDA is quite competitive with many strong baselines including the 2024 one (WTTM). Table 10 and Appendix F (L1005-1018) provides details.
>
> **Improve the interpretability of Fig. 1 \& 2.**
>
> Great suggestion! We plan to split Fig. 1(a) and Fig. 1(b) into separate figures: Fig. 1(a) will be replaced with Fig. 7 in the updated paper to motivate Proxy-FDA in a neat way; Fig. 1(b) will act as a supplementary visualization. Similarly for Fig. 2 , we will focus only on the (Proxy-)FDA loss by replacing Fig. 2(a)\&(c) with Fig. 8 that build things up slowly. While Fig. 2(b) will be moved to Appendix to illustrate the proxy generator architecture separately.

---

> > ### Author Response · Authors · 2024-12-01
> > **Gentle reminder of the discussion deadline**
> >
> > Dear Reviewer 5Vid, thanks again for your insightful feedback! We are sending you this gentle reminder since the rebuttal deadline is approaching. Would you mind letting us know if your concerns have been fully addressed or if you have more questions? In the latter case, we will be happy to follow up. Looking forward to your reply and thanks for your time!

---

### Meta-Review · Area_Chair_t9ip · 2024-12-17

**Metareview:**

This paper presents an algorithm for continual learning. In particular, given a vision foundation model, the task is to fine-tune it consecutively to a series of image classification datasets and the goal is to minimize the forgetting phenomenon, i.e. how the accuracy on the first dataset(s) downgrades throughout the entire fine-tuning procedure. The proposed method involves a regularization technique that uses an explicit structure (proxy) to perform feature distribution alignment. Experiments are performed on several ordered lists of image classification datasets.

This is a borderline paper with an initial rating of 5/5/6/6. One reviewer raised the score from 5 to 6 while others remained unchanged (two of them did not respond). The AC reads the paper and does not find clear drawbacks in the method part. However, the experiments seem insufficient to validate the (strong) effectiveness of the method, because the improvements are mostly marginal. More importantly, the studied setting is of limited interest to the community in the large model era; this concern becomes especially clear given the studied task is standard image classification and some datasets are small and/or toy (e.g. CIFAR10/100 and SVHN). After careful consideration, the AC recommends rejection.

**Additional Comments On Reviewer Discussion:**

The authors provided detailed rebuttals and tried to discuss them with the reviewers. One reviewer raised the score from 5 to 6 and another reviewer kept the score as 6; other two reviewers (5 and 6) did not respond.

---

### Decision · Program_Chairs · 2025-01-22

Reject